# Post-Traumatic Expressions of Aromatase B, Glutamine Synthetase, and Cystathionine-Beta-Synthase in the Cerebellum of Juvenile Chum Salmon, *Oncorhynchus keta*

**DOI:** 10.3390/ijms25063299

**Published:** 2024-03-14

**Authors:** Evgeniya V. Pushchina, Mariya E. Bykova, Anatoly A. Varaksin

**Affiliations:** Zhirmunsky National Scientific Center of Marine Biology, Far Eastern Branch, Russian Academy of Sciences, 690041 Vladivostok, Russia; stykanyova@mail.ru (M.E.B.); anvaraksin@mail.ru (A.A.V.)

**Keywords:** cystathionine-beta-synthase, aromatase B, cerebellum, glutamine synthetase, juvenile chum salmon, *Oncorhynchus keta*, neuroepithelial cells, neuronal stem/progenitor cells

## Abstract

In adult fish, neurogenesis occurs in many areas of the brain, including the cerebellum, with the ratio of newly formed cells relative to the total number of brain cells being several orders of magnitude greater than in mammals. Our study aimed to compare the expressions of aromatase B (AroB), glutamine synthetase (GS), and cystathionine-beta-synthase (CBS) in the cerebellum of intact juvenile chum salmon, *Oncorhynchus keta*. To identify the dynamics that determine the involvement of AroB, GS, and CBS in the cellular mechanisms of regeneration, we performed a comprehensive assessment of the expressions of these molecular markers during a long-term primary traumatic brain injury (TBI) and after a repeated acute TBI to the cerebellum of *O. keta* juveniles. As a result, in intact juveniles, weak or moderate expressions of AroB, GS, and CBS were detected in four cell types, including cells of the neuroepithelial type, migrating, and differentiated cells (graphic abstract, A). At 90 days post injury, local hypercellular areas were found in the molecular layer containing moderately labeled AroB+, GS+, and CBS+ cells of the neuroepithelial type and larger AroB+, GS+, and CBS+ cells (possibly analogous to the reactive glia of mammals); patterns of cells migration and neovascularization were also observed. A repeated TBI caused the number of AroB+, GS+, and CBS+ cells to further increase; an increased intensity of immunolabeling was recorded from all cell types (graphic abstract, C). Thus, the results of this study provide a better understanding of adult neurogenesis in teleost fishes, which is expected to clarify the issue of the reactivation of adult neurogenesis in mammalian species.

## 1. Introduction

Various models of traumatic brain injury (TBI) in teleost fish have been developed [1]. Such injuries can be multidimensional, including several subtypes [1,2] with different neurobiological underpinnings [3,4,5] and several comorbidities [6,7,8]. Transection has been used in a number of studies on the axonal regeneration and recovery of motor function after spinal cord injury [2,3,4]. The second lesion paradigm used to study regeneration after spinal cord injury is based on the amputation of the tail, including the caudal part of the spinal cord [5]. Lesions are applied mechanically, by destroying the dorsal part of the body of the cerebellum [9,10]. The resulting stab wound lesion extends from the pial surface through the dorsal molecular layer to approximately the middle of the granular layer of the cerebellar body [1]. In models of the cerebellum, telencephalon, and tectum TBIs, control brain samples can be obtained either from the area corresponding to the site of the lesion, or from intact or sham-operated animals, or from the contralateral hemisphere in affected animals [1].

Several animal models of TBIs have been used, mainly in rodents such as rats and mice. However, reparative mechanisms in the mammalian brain are very limited, and newly formed neurons do not survive for long periods, probably due to an unsuitable local environment. The striking differences in regenerative properties between the brains of mammals and fish are explained by the remarkable differences in the processes of neurogenesis in adult fish [4,5,11,12]. The results of preliminary studies on the cerebellum in juvenile chum salmon have shown a high degree of conservation of some cerebellar structures compared to those in mammals, as well as similar key neuroanatomical and neurochemical pathways associated with human diseases. Unlike the mammalian cerebellum, that of juvenile chum salmon exhibits intense neurogenesis that can be correlated with highly regenerative properties.

Previous studies have presented and compared the intrinsic and extrinsic determinants of epigenetic regulation underlying both repair and regeneration after TBIs in mammals [13,14,15] and fish [16,17]. A clearer understanding of the molecules and molecular mechanisms of epigenetic regulation that influence the extraordinary individualism in the central nervous systems (CNSs) of adult zebrafish and other animals capable of regeneration may promote/facilitate the development and advancement of treatments for TBI and their clinical application in humans. Primary injury results in an immediate and direct mechanical damage to brain activity, mainly affecting the central gray matter and relatively sparing the white matter, which occurs mainly peripherally [4,18,19]. Primary injury involves the early occurrence of bleeding leading to hypoxia and ischemia, microhemorrhages, or edema near the site of injury [20]. This causes an interruption of nerve transmission as the neurons passing through the injury site become physically damaged and have a reduced myelin thickness [20,21]. Gray matter is considered to be irreversibly damaged within the first hour after injury, whereas white matter is irreversibly damaged within 72 h post injury [20,21]. Worsening may occur within the first 3 h and may continue for at least 24 h [22]. A TBI is followed by several critical abnormalities including hemorrhage, demyelination, edema, and cavitation with the death of axons and neurons, and also by a number of degenerative changes in neural tissues that can precipitate infarction [23,24]. Excitotoxicity, oxidative damage, and ischemia can be caused by increased amounts of glutamate, while Ca^2+^-dependent nitric oxide synthesis can induce secondary spinal cord injury [25]. The increased lipid peroxidation and free radical damage to the cell membrane, as well as additional damage signaling cascades at the site of injury, may ultimately lead to neuronal death after the secondary injury [26]. An initial primary immune response is required to repair the initial damage at the injury site. However, the multifactorial process involving the recruitment of reactive astrocytes, microglia, macrophages, glial progenitors, fibroblasts, and Schwann cells during the secondary injury leads to the formation of persistent glial scarring [27,28]. This glial scar is often impermeable and contains secreted and transmembrane molecules that inhibit axons’ growth [29]. There may also be a gradual spread of the injury to more than one site (syringomyelia) over months or years, often becoming catastrophic [29,30].

The study of the neurogenesis processes in juvenile chum salmon is of certain interest, especially due to the endemism of this species [31]. For the purpose of studying the features of CNS recovery in a commercially valuable species, the chum salmon populations was easily accessible for our team, which allowed us to obtain the required age stages. Also, phylogenetically ancient groups, which include salmonids, are characterized by a high concentration of undifferentiated components both in the matrix areas of the brain and in the parenchyma. In the adult state, chum salmon retain the embryonic state of certain organs or their systems, including the CNS, due to the fetalization processes [32]. It is noteworthy that these processes are superimposed at the stage of active growth, when intensive neurogenesis occurs, which makes this species a convenient model for research. In addition, salmonids do not have the specific features found in other studied fish species, such as electroreception organs or being characterized by a short life cycle. A promising strategy for the development of effective cell replacement therapy, including studies on organisms capable of regeneration, is expected to provide a broader understanding of the restoration of tissues in the adult CNS. Thus, the study of regenerative-competent organisms, in particular juvenile Pacific salmon, can address questions as to how regeneration occurs in species with a delayed embryonic developmental phenomenon (fetalization) and what role the enzymes play that were previously discovered in the neural stem progenitor cells (NSPC) of juvenile salmon, particularly glutamine synthetase (GS) and cystathionine-beta-synthase (CBS) [33], as well as the radial glia (aNSC) of other fishes [12,13].

Estrogens play an important role in the development of the vertebrate brain, having effects on both the activity and morphology of existing neuronal circuits and modulating the neurogenesis in embryos and adult animals [34]. Local aromatization occurs in specific regions of the brain such as the hypothalamus and limbic system. Brain-produced estrogens, known as neuroestrogens, have been found to perform neurotrophic [35] and neuroprotective [36,37] functions. The cyp19a1 gene encodes aromatase, and in most vertebrates, including mammals, it is expressed in the brain. However, teleost fish have evolved to have two distinct cyp19a1 genes, *cyp19a1a* and *cyp19a1b* [38]. In teleost fish, both *cyp19a1a* and *cyp19a1b* are expressed in the brain, but their expression patterns and regulation can vary depending on the species, sex, and developmental stage [39]. Aromatase A (encoded by *cyp19a1a*) is generally expressed at higher levels during early development and in mature gonads, whereas aromatase B (encoded by *cyp19a1b*) is more dominant in the adult fish brain [40]. The presence of two different *cyp19a1* genes in teleost fish provides a more fine-tuned regulation and control of estrogen production in the brain. The *cyp19a1* genes are regulated by brain-specific promoters that are responsible for initiating gene expression specifically in the brain [41]. Aromatase B (AroB) is an enzyme that is expressed in the RG cells of adult fish [39,42] and plays a key role in the production of estrogens in *Danio rerio* [42]. Overall, the expression of cyp19a1 genes in the brains of vertebrates, including the two different genes in teleost fish, is crucial for the estrogen production and regulation of various physiological processes such as breeding, behavior, and neurodevelopment [42,43].

According to the results of comparative ontogenetic studies, the AroB expression in *D. rerio* embryos is very low and can be detected only via quantitative polymerase chain reaction Q-PCR [44]. In *D. rerio* embryos, the AroB expression increases slowly and becomes detectable in transgenic animals or via immunohistochemistry 5–7 days later [45]. Radial glia cells (RGCs) express aromatase B and act as precursors only in adult *D. rerio* [46,47,48,49]. Estrogens are suggested to play a key role in the brain neurogenesis and recovery in teleost fish, although data on them still remain controversial [50,51]. In particular, an inhibitory effect of estrogens and estrogen receptors on adult *D. rerio* neurogenesis, but no stimulatory effect on cell proliferation, has been found, which disagrees with the results of studies on mammals [48]. The involvement of neuroestrogens (AroB) in the post-traumatic repair processes of brain TBIs is also unknown.

Another marker found in the NSPCs of embryonic and adult types in the fish brain is glutamine synthetase. Previously, GS has been detected in the intact cerebellum and after an acute traumatic injury in juvenile *Oncorhynchus masou* [33]. In contrast to other fish species [52,53,54], GS in the cerebellum of *O. masou* juveniles is predominantly localized both in neuroepithelial cells (NECs) and in RGCs [33]. GS is a marker of mature glial cells in the cat shark retina [55,56]. In young sharks, GS has been found in tanycytes (ependymal cells), which is consistent with the results obtained for adult mammalian brains [57,58,59]. In adult amphibians and teleosts, cells with the RGC morphology have appeared to express glial markers, including glial fibrillar acidic protein (GFAP) and GS. However, in mammals, these markers are expressed in RGCs but not in NECs, which provides differentiation between the two cell types. It is worth noting that these markers are also expressed in differentiated glial cells such as astroglia and ependymal cells [60]. Therefore, in both mammals and non-mammalian vertebrates, similar markers are expressed by different types of progenitor cells, as well as by different types of differentiated glial cells [60].

This study aimed to compare the patterns of the expressions of aromatase B, glutamine synthetase, and cystathionine-beta-synthase in an intact cerebellum of juvenile chum salmon, *O. keta*. To identify the dynamics that determine the involvement of AroB, GS, and CBS in the cellular mechanisms of regeneration, we assessed the expressions of these molecular markers during a long-term primary traumatic brain injury (TBI) and after a repeated acute TBI to a juvenile *O. keta* cerebellum.

## 2. Results

Previously, cells of types 1–4 were identified in the cerebellum of juvenile masu salmon, *Oncorhynchus masou*, in accordance with their morpho-functional, topographic and proliferative characteristics [33]. Previous studies provided a detailed rationale for the need to identify these cell types and indicated that type 1 cells correspond to NSPCs [33]. Type 2 cells are elongated, larger than type 1 cells, and probably represent the pre-proliferation stage (type 1 cells before division), and may also correspond to mammalian astrocytes [59]. Type 3 cells are cells of the ganglionic layer and are represented by two populations: Purkinje cells and eurydendroid cells (EDCs). Type 4 cells are a population of elongated migratory cells involved in the reparative process and capable of migrating toward the injury zone.

### 2.1. Aromatase B Expression in the Intact Cerebellum of Juvenile O. keta

An AroB immunohistochemistry (IHC) labeling in the dorsal zone (DZ) of the intact *O. keta* cerebellum revealed small, intensely/moderately labeled type 1 cells that formed small bands in the superficial part of the molecular layer (ML) (Figure 1A, Table 1). Weakly/moderately labeled elongated cells were detected deep in the ML of type 4 cells that were arranged into radially oriented strands directed toward the dorsal matrix zone (DMZ) (Figure 1A, Table 1). In some cases, cells of this type formed multidirectional clusters (Figure 1A, red inset, Table 1). Single, intensely labeled type 1 cells were found at the border between the molecular and ganglionic layers (Figure 1A).

In the lateral zone (LZ), clusters of labeled type 1 cells were found in the superficial and subsurface zones (Figure 1B, black inset, Table 1). Single, intensely labeled type 1 cells were localized in the granular layer (Figure 1B, Table 1). Single, intensely labeled type 1 cells and clusters of Aro+ cells in the granular layer (Grl) were detected in the parenchymal part of the basal zone (BZ) (Figure 1C, blue inset, Table 1).

In the area of granular eminence (Gr em), the number of intensely labeled type 1 cells increased substantially (Figure 1D, Table 1). Such cells formed clusters or were located separately deep in the granular layer (Figure 1D, black inset, Table 1). In the latter, most Aro+ cells were represented by moderately labeled type 1 cells that formed ribbon-like clusters or were arranged singly (Figure 1E, pink inset, Table 1).

A comparative analysis of the distribution of Aro+ cells in various anatomical parts of the intact juvenile *O. keta* cerebellum showed a minimum number of cells in the BZ and a maximum number in the granular layer (Figure 1F). Significant intergroup differences were found between different regions of the cerebellum (Figure 1F).

### 2.2. Aromatase B Expression at 90 Days after Injury to the Juvenile O. keta Cerebellum

Hypercellular aggregations with intensely labeled type 1 cells (4000 µm^2^) were detected in the DZ at 90 days after a puncture injury to the juvenile *O. keta* cerebellum (Figure 2A, black inset, Table 1). In the DMZ, single, intensely labeled type 1 cells and their clusters were identified, which were absent in intact animals (Figure 2A, green inset, Table 1). Single Aro+ radial glia (RG) fibers were found at the base of the DMZ (Figure 2A, green inset, Table 1). Numerous clusters of immunopositive type 1 cells were detected in the dorsal part of the granular layer (Figure 2A, Table 1).

In the LZ, the number of intensely labeled type 1 cells, forming hypercellular clusters with an area of up to 7000 µm^2^, also increased (Figure 2B, black inset, Table 1).

In the BZ, along with separate, intensely labeled type 1 cells, separate, intensely labeled type 2 cells were also present deep in the molecular layer (Figure 2C, Table 1). At 3 months, the distribution patterns of intensely labeled type 4 cells, absent in intact animals, were also characteristic of the BZ (Figure 2C, red inset, Table 1). Numerous, intensely labeled type 1 and type 2 cells were identified in the granular layer (Figure 2C, Table 1). Ganglionic layer cells were immunonegative (Figure 2C, black inset, Table 1).

In the granular layer, numerous intensely labeled type 1–2 cells were typical and formed diffuse clusters or dense local conglomerates of heterogeneous morphological compositions (Figure 2D, Table 1). Ribbon-like groups of uniform migrating immunopositive cells were frequently observed in the diffuse clusters (Figure 2D, blue and red insets, Table 1). In local dense heterogeneous conglomerates, intensely labeled Aro+ cells co-occurred with poorly labeled or Aro-negative type 1 cells (Figure 2D, black inset, Table 1). Figure 2E shows the area of injury at 3 months post injury. In this area, a post-traumatic scar had formed as a result of regeneration, and a large number of intensely and moderately labeled Aro+ cells of types 1 and 2 were visualized (Figure 2E, red inset). In the Gr em, large aggregations of type 1 cells were found, located in the superficial and parenchymal layers (Figure 2F, black and blue insets, Table 1). Deep in the Gr em, cells of four types were identified along the guides (Figure 2F, red inset, Table 1).

The results of the comparative analysis showed that at 90 days post injury, there was a significant increase in the number of immunopositive cells in the DZ, LZ, and BZ compared to intact animals (*p* ≤ 0.01). In the Gr em, the number of immunopositive cells also significantly increased (*p* ≤ 0.05) (Figure 2G).

### 2.3. Expression of Aromatase B during Repeated Injury (90 Days + 7 Days) in the Juvenile O. keta Cerebellum

As a result of acute repeated injury to the juvenile *O. keta* cerebellum, a significant increase in the number of intensely labeled, Aro+ type 1 cells, forming hypercellular clusters, was observed in the DZ (Figure 3A, red inset, Table 1). In the basal part of the molecular layer, patterns of Aro+ type 1 cells located along the guides were typical (Figure 3A, black inset, Table 1). In some cases, moderate aromatase B immunopositivity was detected in EDCs (Figure 3A, green inset, Table 1).

In the LZ, the acute repeated injury induced AroB immunopositivity in hypercellular foci of up to 9000 µm^2^ in size (Figure 3B, red inset). In the ganglionic layer, AroB immunopositivity was detected in Purkinje cells (PC) (Figure 3B, blue inset, Table 1). Radially oriented type 4 cells were identified in the molecular layer (Figure 3B, Table 1). Numerous clusters of intensely labeled type 1 cells were detected in the granular layer (Figure 3B, Table 1). Dense and diffuse clusters of intensely labeled type 1 and 2 cells were detected in the lower third of the BZ molecular layer (Figure 3C, red and black insets, Table 1).

Single intensely/moderately labeled type 1 and 2 cells were observed in the Gr em area (Figure 3D, red inset, Table 1). Single cells were present in the superficial layers (Figure 3D, red inset, Table 1). In the deep parenchymal layers, heterogeneous patterns of migration of type 1 and 2 cells were identified (Figure 3D, Table 1). In areas of the granular layer remote from the injury area, the distribution patterns of diffuse clusters of Aro+ type 1 cells were most typical (Figure 3E, Table 1). In the granular layer, single Aro+ cells were detected in the region adjacent to the site of injury (Figure 3F, green dotted line, Table 1), with mixed denser aggregations of Aro+ cells extending deep and laterally from it (Figure 3F, red and blue insets, Table 1).

Comparative analyses of the distributions of Aro+ cells in intact animals at 90 days after the primary injury and at 7 days after the repeated injury are presented in Figure 3G. The most intense dynamics were observed in the DZ and BZ, where the number of Aro+ cells increased multifold as a result of primary and repeated injuries compared to the control (*p* ≤ 0.01). In the LZ and Gr em, the number of cells significantly increased both after the primary injury (*p* ≤ 0.05) and after the repeated injury (*p* ≤ 0.01) compared to the intact animals. In the granular layer, the number of cells increased significantly (*p* ≤ 0. 05).

### 2.4. Expression of Glutamine Synthetase in the Intact Juvenile O. keta Cerebellum

The IHC labeling for GS in the dorsal part of the juvenile *O. keta* cerebellum revealed intensely/moderately labeled type 1 cells, single or forming small clusters, in the granular layer (Figure 4A, red inset). The molecular layer was dominated by single, intensely labeled type 1 and 2 cells. At the border between the molecular and granular layers, there were guides containing moderately labeled cells (Figure 4A, Table 2). In the LZ, intensely labeled type 1 and 2 cells also dominated in the parenchyma of the molecular layer and deep in the granular layer. Superficially located guides contained weakly or moderately labeled cells or fragments of radial fibers (Figure 4B, Table 2).

Moderate GS activity was observed in some elongated EDCs (Figure 4B, Table 2). In the BZ, denser distribution patterns of guides were recorded from the molecular layer and at the border between the ganglionic and granular layers containing moderately labeled cells (Figure 4C, Table 1, black square). In the lower third of the molecular layer, GS+ type 1 and 2 cells were quite abundant (Figure 4C, Table 2). Separately located, intensely labeled type 1 and 2 cells were identified in the gaps between the guides in the BZ (Figure 4D). The granular layer was characterized by a diffuse distribution of GS+ type 1 and 2 cells without pronounced clusters (Figure 4E, black inset, Table 2). In the Gr em area, small clusters of intensely labeled type 1 and 2 cells and moderately labeled type 4 cells were identified in the superficial layers (Figure 4F, Table 1).

The results of the comparative analysis of GS+ cells indicated the predominance of type 1 cells over type 2 cells in all the cerebellum regions (Figure 4G). Intergroup differences in the number of GS+ cells were found in the Grl and in the Gr em compared to the granular layer (*p* ≤ 0. 05).

### 2.5. GS Expression at 90 Days after the Injury to the Juvenile O. keta Cerebellum

A long-term monitoring in the DZ of the cerebellum revealed large hypercellular foci up to 15,000 µm^2^ (Figure 5A, red inset, Table 2) containing intensely labeled type 1 cells. In the superficial layers of the molecular layer, intensely labeled type 1 and 2 cells were detected (Figure 5A, Table 2). Hypercellular clusters with complex geometries up to 19,000 µm^2^ in size, which included guides containing cells of moderately labeled cells, were also found in the LZ (Figure 5B,C, Table 2). Similar structures were identified at the border between the granular and ganglionic layers (Figure 5B, Table 2).

In the cerebellar BZ, in the granular layer region, there were post-traumatic areas devoid of cells (Figure 5D, red inset, Table 2), along which single, intensely immunolabeled type 1 and 2 cells and radial guides were located (Figure 5D, red inset, Table 2). Numerous guides and their fragments, containing type 1 and 2 cells (Figure 5E, black inset, Table 2), were present in the granular layer (Figure 5E, black inset, Table 2).

Large post-regenerative cellular patterns containing dense aggregations of type 1 and 2 cells and also single, intensely labeled type 1 and 2 cells (Figure 5F, black inset, Table 2) were observed in the granular layer of the DZ. Intensely labeled diffuse aggregations of type 1 cells (Figure 5G, black inset) were detected in the superficial layers of the Gr em.

The results of the comparative analysis of distribution of type 1 and type 2 cells after the long-term primary injury showed a significant increase in type 1 cells in the DZ, LZ, Gr em, and the granular layer (*p* ≤ 0.01) and a less pronounced increase in the BZ (*p* ≤ 0.05) (Figure 5H).

### 2.6. GS Expression upon Repeated Injury (90 Days + 7 Days) in the Juvenile O. keta Cerebellum

The repeated acute injury to the DZ of the juvenile *O. keta* cerebellum caused numerous patterns of intensely labeled type 1 cells to appear in the superficial regions of the molecular layer (Figure 6A, blue inset, Table 2), as well as linear, radially oriented patterns of the migrations of such cells (Figure 6A, red inset, Table 2), and numerous single intensely labeled type 1 and 2 cells within the molecular layer (Figure 6A, Table 2). Similar patterns were observed at the border between the ganglionic and granular layers (Figure 6A, Table 2).

In the cerebellum LZ, in the superficial and deeper regions of the molecular layer, the distribution density of intensely labeled type 1 and 2 cells increased significantly (Figure 6B, red inset, Table 2). In cells of the ganglion layer, there was a moderate or intense level of GS activity, at which cell bodies and proximal parts of their dendrites were labeled (Figure 6B, Table 2). Intensely labeled type 1 and 2 cells were found among cells of the ganglionic layer (Figure 6B). Dense clusters of intensely labeled type 1 and 2 cells in the molecular layer (Figure 6C, red inset, Table 2), as well as radially migrating type 4 cells, were observed in the BZ. In the granular layer, radially directed fibers, and single, intensely labeled type 1 and 2 cells (Figure 6E, Table 2) were found. In the Gr em, extensive dense aggregations of intensely labeled type 1 cells were observed in the superficial (Figure 6F, red inset, Table 2) and deeper layers (Figure 6F, red inset).

A comparative study of the distribution of GS+ type 1 and 2 cells during the long-term primary and acute secondary injury to the cerebellum showed significant increases in the number of GS+ type 1 cells (*p* ≤ 0.01) in all the examined areas as a result of both the primary and secondary injuries.

### 2.7. CBS Expression in the Intact Juvenile O. keta Cerebellum

The IHC labeling for CBS in the DZ of the intact cerebellum of *O. keta* juveniles revealed weakly or moderately labeled type 1 and 2 cells, (Figure 7A,B, Table 3). In the molecular layer, small diffuse aggregations of type 1 cells were detected (Figure 7B, black inset). Similar patterns were found in the LZ (Figure 7B, Table 3).

In the BZ, the number of intensely labeled type 1 cells in the superficial layers was higher (Figure 7C, black inset, Table 3). Chains of CBS+ cells were observed deep in the molecular layer (Figure 7C, red inset, Table 3). In the granular layer, diffusely localized type 1 and 2 cells with moderate CBS labeling were detected (Figure 7D, pink and blue insets, Table 3). In the Gr em, the diffuse pattern of the distribution of moderately labeled type 1 and 2 cells was also observed (Figure 7E, Table 3).

A comparative analysis of the distribution of CBS+ type 1 and 2 cells in the cerebellum of *O. keta* juveniles showed a significant dominance of type 1 cells in all the studied areas (Figure 7F).

### 2.8. Expression of CBS at 90 Days Post Injury to the Juvenile O. keta Cerebellum

As a result of the long-term monitoring of the primary injury, in the DZ of the juvenile *O. keta* cerebellum, post-traumatic hypercellular zones of up to 7500 µm^2^ were found that contained type 1 cells with intense labeling (Figure 8A, red inset, Table 3). In the basal part of the molecular layer, at the border with the granular layer, patterns of intensely labeled type 1 cells located between projections of type 3 cells were often detected (Figure 8A, green inset, Table 3). Similar patterns were also observed at the border between the granular and ganglionic layers (Figure 8A, Table 3).

Numerous morphological post-traumatic changes in the cerebellar nervous tissue, in particular, local concentric accumulations of intensely labeled type 1 cells including cell-free cavities, were found in the LZ (Figure 8B, red inset, Table 3). In some cases, radially directed tracks of cells were identified in the molecular layer (Figure 8B, Table 3). Large, moderately labeled type 2 cells were identified in the granular layer of the LZ (Figure 8B, Table 3). Intense labeling was also observed in cells of the ganglion layer (Figure 8B, red inset, Table 3).

In the BZ, there were patterns of intensely labeled type 1 cells (Figure 8C, red inset, Table 3), radially oriented cells (Figure 8C, pink inset). In the DZ, extensive post-traumatic lacunae, devoid of cells and surrounded by scattered aggregations of CBS+ type 1 and 2 cells (Figure 8D, dotted line, Table 3), as well as heterogeneous moderately labeled CBS+ larger type 4 cells, were observed (Figure 8D, red inset, Table 3). In the Gr em, there was an accumulation of densely labeled type 1 and 2 cells in the superficial layers (Figure 8E, black inset, Table 3) and larger cells in the deeper layers (Figure 8E, black inset).

A comparative analysis of the distribution of type 1 and type 2 CBS+ cells in intact animals and as a result of long-term primary injury showed that the number of type 1 cells significantly increased in the LD, LZ, Gr em, and in the granular layer (*p* ≤ 0. 05). In the BZ, no significant increase in the number of cells was found after the primary injury (Figure 8F).

### 2.9. CBS Expression upon Repeated Injury (90 Days + 7 Days) in the Juvenile O. keta Cerebellum

As a result of repeated acute injury to the juvenile *O. keta* cerebellum, patterns of an increased number of CBS+ cells of types 1 and 2 appeared in the granular layer (Figure 9A, yellow inset, Table 3). In the apical part of the cerebellum, extensive, superficially localized clusters of intensely labeled type 1 cells (Figure 9B, red inset) were observed. Large, intensely labeled infiltrates containing type 1 and 2 cells were detected in the LZ (Figure 9C, red inset).

In the BZ at the border between the granular and ganglionic layers, the number of single CBS+ cells of types 1 and 2, as well as their clusters, significantly increased (Figure 9D, red inset). Similar aggregations were identified in the granular layer in the area of damage and contained CBS+ type 1 and 2 cells with different distribution densities, forming local immunopositive clusters (Figure 9E). In the Gr em, superficially located type 1 cells were particularly numerous (Figure 9F, black and red inset); less dense aggregations of intensely labeled type 1 cells were found deeper in the granular layer.

A comparative analysis of the distribution of type 1 and type 2 CBS+ cells in intact animals after the primary and repeated injuries showed that the number of type 1 cells increased noticeably in the DZ, LZ, Gr em, and in the granular layer after the primary injury (*p* ≤ 0. 05) and significantly after the secondary injury (*p* ≤ 0.01). In the BZ, there was no significant increase in the number of cells after the primary injury, while a significant increase in the number of cells was observed after the repeated injury (*p* ≤ 0.05) (Figure 9G).

## 3. Discussion

### 3.1. Aromatase B Expression in Adult Neurogenesis

The expression and regulation of aromatase have been studied for various fish species [39,44,45,47,61,62,63,64]. Various methods, including mRNA in situ hybridization, immunohistochemistry using antibodies specific for aromatase B in *D. rerio*, and GFP expression driven by the *cyp19a1b* promoter, have been used to confirm the presence of aromatase B expression in *D. rerio* RGCs [34]. RGCs are widely distributed in various areas of the fish brain, including the telencephalon, preoptic region, thalamus, hypothalamus, *tectum opticum*, and *torus semicircularis*; however, there is a lack of data on the RG expression in the cerebellum of *D. rerio* [34]. It is noteworthy that in *D. rerio*, *cyp19a1b* mRNA is indicated not only in the somata of cells localizing the ventricles, but also in the RG and terminal peduncles along the meninges, where it accumulates [65].

In our studies, several types of cells expressing AroB in intact conditions were identified in the juvenile *O. keta* cerebellum (Table 1). Unlike other fish species, AroB in juvenile *O. keta* was mainly found in the somata of small, type 1 cells located in all the neuroanatomical zones of the cerebellum. The maximum number of Aro+ cells was found in the granular layer of the cerebellar body and Gr em (Figure 10) and a somewhat lower number in the molecular layer of the dorsal and lateral regions of the cerebellar body (Figure 10), while the number of immunopositive cells in the BZ was minimal.

Thus, in the intact *O. keta* juveniles, we did not find Aro+ RGCs, in contrast to other studied fish species [40,46,47,48,65]. We suggest that this feature may be associated with the age characteristics of the studied fish (at 16 months of age), which corresponds to the juvenile stage of Pacific salmon development in a state of active growth. In *D. rerio* and some other fish species, aromatase B activity and expression in the brain have been shown to remain low during the embryonic period and begin to increase during the maturation, coinciding in time with increased levels of circulating sex steroids such as estradiol and testosterone [66,67]. It has also been found that in *D. rerio* embryos, the aromatase B expression can be significantly increased when exposed to any estrogen compound, even at very low doses. This indicates that the aromatase B expression is highly dependent on estrogen stimulation during embryonic [62,66] and, possibly, postembryonic development.

Previous studies showed that the aromatase B expression in *D. rerio* embryos is very low and can be detected using only Q-PCR [44]. An increase in the expression of *cyp19a1b* mRNA is observed between 24 and 48 h post fertilization in parallel with the expression of the estrogen receptor [45,61,63]. In *D. rerio* embryos, the aromatase B expression increases slowly and becomes detectable in transgenic animals or via immunohistochemistry after approximately 5–7 days [45]. In adult fish, RGCs express aromatase B and act as precursors [46,47,48], which evidently suggests a functional relationship between estrogen synthesis and the proliferation of these cells [45].

The relative aromatase B levels detected both through in situ hybridization and Q-PCR indicate that the most abundant expression of aromatase in *D. rerio* is characteristic of the subpallial regions of the telencephalon along the midline of the ventricle [41]. In addition, significant expression was observed in the diencephalic regions of the preoptic region, thalamus, and hypothalamus associated with the third ventricle. This expression pattern is consistent with that reported for other studied teleosts [41] and is also present in the olfactory bulbs and pallial ventricular regions of the dorsal telencephalon. However, caudal brain regions differ among fish species [39,42,68,69].

Thus, in the juvenile *O. keta* cerebellum, which is not a sensory zone of the brain, in contrast to the hypophysiotropic regions of the preoptic region and the hypothalamus of the Atlantic salmon [67,70], a relatively low content of AroB was found in type 1 cells representing the population of NSPCs and located in characteristic zones with proliferative activity. In addition to type 1 cells, a few larger type 2 cells and a population of migrating type 4 cells were found in the cerebellum of intact juvenile *O. keta*. The detection of aromatase B activity in these cell types may be related to their estrogenic modulation. In particular, the processes of the growth, differentiation, and migration of cells during constitutive neurogenesis can be regulated by estrogens, which is consistent with data on other species [41,43,71,72].

In a study on the plainfin midshipman *Porichthys notatus*, antibodies were used to detect aromatase B. These antibodies were also used in a study on the rainbow trout *Oncorhynchus mykiss*, where abundant aromatase B-immunoreactive enzymes were found in various lobes of the pituitary gland, cells bounding the ventricles in the telencephalon, and ventral diencephalic cells expressed high levels of aromatase B in the preoptic region and hypothalamus [73].

The aromatase B positive cells in both species had similar characteristics. They had small ovoid nuclei located along the ventricles, a short terminal stalk directed toward the ventricle, and a long radial process terminating on the pial surface. In rainbow trout, these cells were also observed in the *torus semicircularis* and the *tectum opticum*. In the *tectum opticum*, the cell bodies were located in the ependymal wall, and their long processes penetrated the parenchyma through all layers of the tectum [73].

However, there was a contrast in the data between rainbow trout and plainfin midshipman. While aromatase B was detected in the tectum of the former, it was not detected in the tectum of the latter [42].

The distribution of aromatase in fish is also associated with sexual dimorphism [34]. For example, in *D. rerio*, sexual dimorphism is weakly expressed and does not affect aromatase expression, while in the medaka *Oryzias latipes*, having pronounced sexual dimorphism, females showed a higher expression in the ventricular layer of the *tectum opticum* compared to males [49]. On the other hand, studies on the European sea bass (*D. labrax* L.) did not reveal significant sexual dimorphism in aromatase activity [62]. Thus, sexual dimorphism is more pronounced in the medaka *O. latipes*, a species with a strong genetic determination of sex, in contrast to the zebrafish *D. rerio* and the sea bass *D. labrax L.* [34].

The expression and regulation of aromatase B in the brain of adult teleost fish have unique characteristics compared to those in other vertebrates. The high aromatase activity in the brain of teleost fish is primarily caused by the strong expression of the *cyp19a1b* gene [39,74]. This gene is selectively expressed in RGCs in numerous fish species [39,42,45,75]. In other vertebrates, RGCs act as progenitors during the embryonic development but are transformed into astrocytes, or other cell types, or the so-called B cells after the embryonic period [39]. However, other studies suggest that in teleost fish, radial glia cells (RGCs) continue to exist in different regions of the brain and play a role in the brain’s ability to grow and develop throughout adulthood [74]. In particular, studies on *D. rerio* [47,49] and the pejerrey *Odontesthes bonariensis* [63] have demonstrated that RGCs, some of which produce aromatase, maintain their ability to generate new neurons and act as progenitor cells throughout the lifespan of the adult individual. In the Japanese eel *Anguilla japonica* [40] and in the plainfin midshipman *P. notatus* [42], the expression of aromatase was found to be limited by RGCs, as in some other bony fish species [39,45,47,48,64]. The expression of aromatase B has primarily been observed in neurons in mammals and birds [36,76,77,78]. However, more recent studies have also identified aromatase expression in astrocytes [79,80]. In the case of zebra finches, the expression of aromatase in RGCs was found to occur specifically after a brain injury [81].

### 3.2. Aromatase B Expression Post Injury

The estrogen modulation of brain functions is of great importance for the treatment of ischemia, trauma, inflammation, and neurodegeneration associated with neurological disorders and aging [82]. Although aromatase is known to be predominantly expressed in mammalian and avian neurons, under certain circumstances it can also be expressed in astrocyte cells, especially under conditions of brain injury. However, in many teleosts [42,44,48,61,62,63], RGCs strongly express aromatase, which raises questions as to when and why this feature appeared in the course of evolution. It is still unclear whether this particular event is unique to fish or if it also occurs in tetrapods [34].

The long-term monitoring after the primary injury to the juvenile *O. keta* cerebellum showed an increase in type 1 Aro+ cells in the MLs of the dorsal, lateral, and basal regions of the cerebellar body (Figure 2F). These Aro-producing cells, which we identified as part of hypercellular clusters, more closely correspond to neuronal precursors, and, in our opinion, their appearance is associated with traumatic injury. However, we suggest that the aggregations of small Aro+ cells in the chum salmon cerebellum are not analogous to Aro+-reactive astrocytes in mammals [83,84] and birds [81] that appear after injury.

It has been shown that mechanical or chemical damage activates an aromatase expression in reactive astrocytes surrounding the affected area or in RG cells facing the damaged area, as is observed in birds [80,81,84,85,86,87]. The results of our observations have shown that in a long post-injury period (90 days) in the juvenile *O. keta* cerebellum, it is the type 1 cells without processes that increase in number, while the proportion of larger, type 2 Aro+ cells (which may be analogous to reactive mammalian astrocytes) is insignificant. Thus, our results confirm that the aromatase expression in astroglial and NSC/NPC cells in chum salmon may be part of the mechanisms supporting brain recovery after injury [85,86,87,88].

Studies on *D. rerio* using the proliferation markers BrdU, proliferating cell nuclear antigen (PCNA), and aromatase B have shown that Aro+ RG cells actively divide with the formation of new cells [47]. These newborn cells can undergo further division, migrate along the radial processes, and eventually differentiate into neurons [46,47,49]. Since aromatase B is specifically expressed in brain progenitor cells, this actually suggests some role of estrogen in constitutive neurogenesis. However, data on *D. rerio* indicate that aromatase B does not seem to be activated in proliferating cells near the damaged area, which corresponds to the low content of Aro+ cells in the granular layer and granular eminences of juvenile *O. keta*.

As has been shown previously, estradiol has an inhibitory effect on cell proliferation under physiological conditions, and the treatment of *D. rerio* fish with estradiol does not seem to affect cell proliferation after injury [11,46,50,89]. Another noteworthy finding is the detection of aromatase B immunopositivity in the projection cells of the ganglionic layer of chum salmon after primary injury. This feature distinguishes juvenile *O. keta* from other fish species, in particular *D. rerio* [61], but it is consistent with data on the brains of mammals and birds, where aromatase is expressed in neurons [90].

In the rainbow trout *O. mykiss*, *D. rerio*, pejerrey *O. bonariensis*, Japanese eel *A. japonica*, and medaka *O. latipes*, the aromatase expressions are limited to RGCs [39,40,61,62,65]. A possible explanation for this is the broader inclusion of estrogenic anti-inflammatory pathways activated after cerebellar injury [91]. This assumption is consistent with the results of studies on the goldfish *C. auratus* [92] and confirms the hypothesis of endogenous estrogens that have a neuroprotective effect and increase neuronal plasticity as a result of injury.

In the case of repeated injury in an acute period, we observed a further increase in the number of type 1 Aro+ cells in the DZ, LZ, and in the granular eminences (Figure 3A–F). This indicates that repeated injury to the *O. keta* juvenile cerebellum does not decrease the activity of NSPCs, and the number of such cells, including forming reactive neurogenic niches, continues to increase (Figure 3A–D). Furthermore, after the repeated injury, the aromatase immunopositivity in almost all areas of the cerebellum was detected in the PCs and EDCs, which indicates an increased neuronal plasticity and, possibly, an increasing neuroprotective anti-inflammatory effect of AroB in the repeated post-injury period.

Estrogens are considered “neuroprotectors” in the adult brain because they have protective effects by preventing excitotoxicity, inflammation and oxidative damage; inhibiting apoptosis; promoting neuronal survival; regulating dendritic remodeling, synaptogenesis, steroidogenesis; and modulating various neurotransmitters such as acetylcholine, dopamine, serotonin, glutamate, and gamma-aminobutyric acid (GABA) [35,80,93,94,95,96,97]. It has been shown that a chronic decrease in endogenous estrogens via aromatase inhibition through fadrozole in vivo leads to changes in the expressions of different genes and pathways involved in the neuroendocrine function and neuronal plasticity in the brain of the goldfish *C. auratus* [92].

Responses to primary and repeated damage to the cerebellum in chum salmon activate numerous signaling pathways. Components of these pathways are excellent candidates to be inducers of an “astrocyte-like” response in the brain of a juvenile *O. keta*, similar to those in the mammalian brain. Eventually, estradiol from Aro+ cells exert paracrine neuroprotective effects through the potential inhibition of inflammatory pathways. These results indicate a new significance for neuronal aromatization as a mechanism against the development of neuroinflammation. Furthermore, this supports the assumption that the central estrogen supply is a potent neuroprotector in the *O. keta* brain.

Our results are further supported by studies on songbirds, where exposure to kainic acid resulted in an increase in the aromatase expression in astroglia [93]. Similar results were obtained in other models of nervous system injury, including TBI and ischemic stroke, where the increased activity and expression of aromatase was found in reactive astrocytes [96,97]. These studies showed that a wide range of neural lesions can induce an aromatase expression in astroglia and other cell types, suggesting an evolutionarily conserved response to various forms of injury.

### 3.3. Expression of Glutamine Synthetase in Adult Neurogenesis

GS is a CNS enzyme that plays a key role in the metabolism of neurons that express glutamate [58,89,90,91,92,93,94,95,96,97,98,99,100]. Glutamate is taken up by glia and converted by GS to neutral glutamine to prevent neurotoxicity [58].

Previously, GS was detected in the intact *O. masou* cerebellum and after acute traumatic injury [33]. Unlike in other fish species [52,53,54], the localization of GS in the juvenile *O. masou* cerebellum was predominantly detected both in NE type cells and in RGs [33]. GS is a marker of mature glial cells in the cat shark retina [55,56]. The expression of GS in the ependymal cells of young sharks is consistent with the expression of GS in the ependymal cells of adult human and mouse brains [57,58]. This finding is also consistent with previous studies that found GS-positive cells with radial glial cell (RGC) morphology in the brains of adult amphibians [101] and teleosts [52,53,54]. In mammals, glial markers such as GFAP and GS are expressed in RGCs but not in neural stem cells referred to as neural epithelial cells (NECs). However, these markers are also expressed in differentiated glial cells like astroglia and ependymal cells [60]. Therefore, in mammals and other non-mammalian vertebrates, different types of progenitor cells can express similar markers that are shared with various types of differentiated glial cells [52].

According to the results of this study, the proportion of type 1 and type 2 cells in all neuroanatomical regions of the *O. keta* cerebellum in the ML and GrL showed a significant predominance of type 1 cells, which corresponds to the data on *O. masou* [33]. However, the number of type 2 cells within the molecular layer and heterogeneous hypercellular aggregations in *O. keta* varied, being at a maximum in the BZ and at a minimum in the DZ (Figure 4G). In the BZ, similar distribution patterns of type 2 cells have been previously found in *O. masou*, but the minimum number of cells, in contrast to that in *O. keta*, was recorded from the LZ [33]. In the granular layer of *O. keta*, similar patterns of cell distribution were observed, and the number of type 1 cells in the molecular layer and in hypercellular clusters was five times higher than the number of type 2 cells, while in *O. masou*, it was only 2–2.5 times higher. In *O. keta*, hypercellular GS+ regions contained fragments of neuroepithelial structures and a background relative to the hypocellular stroma of the molecular layer. Type 3 cells in all neuroanatomical zones of the *O. keta* cerebellum were represented by a limited population compared to type 1 and 2 cells. Thus, GS+ cells in the cerebellum of *O. keta* juveniles, as well as *O. masou*, are mainly located in the SVZ and PZ of the molecular layer, granular eminences, and in the thickness of the granular layer. A morphological assessment of these mature constitutive neurogenic niches (CNNs) cells showed distinct morphological features. In particular, they were represented by a heterogeneous population of cells forming hypercellular areas with signs of external vascularization, which, to a lesser extent, could correspond to a population of glia-like cells involved in glutamate metabolism in the ganglionic layer.

The data obtained on chum salmon are consistent with the results on zebrafish whose brain was found to have specific cellular and molecular phenotypes of NSPCs [102]. A study on the *D. rerio* model demonstrated that the cells and molecular heterogeneity of NSCs may be related with the regulation of tissue homeostasis and regenerative plasticity of different populations of NSCs during the postembryonic development in *D. rerio* [103]. Being the main morpho-functional units in the construction of the CNS, neuroepithelial and glial NSCs are modulated by external morphogenetic signals, with the activations of certain genetic cascades and the expressions of specific regulatory genes [74,103]. Due to various receptor–transmitter interactions, cell lines in the composition of NE and glial progenitors determine the constitutive growth of the brain in adulthood. However, the specifics of the regulatory impact and the time frames in which the influences of various morphogenetic factors and neurochemical signals on postembryonic neurogenesis in different fish are activated remain completely unstudied. Thus, heterogeneous NE and RG phenotypes of NSPCs in fish and amphibians persists throughout life, playing special roles in the embryonic development of the CNS, while also having stimulating effects on the constitutive neurogenesis in adults [102].

There is increasingly more evidence that the proportion of NE and RG precursors in the brain varies significantly between different fish species [34]. Studies on the medaka *O. latipes* [104] and the zebrafish *D. rerio* showed different ratios of NE precursors in the telencephalon and in the optic tectum, of which its functional significance is currently unclear. However, in intact *O. keta* juveniles, no GS+ RG cells have been found in the cerebellum. This high neural stem cell (NSC) potential in NE cells is due to their origin from embryonic NSCs during early stages of CNS development. NE cells have the ability to differentiate into different types of neurons and a diverse population of glial cells. This multipotency is a key characteristic of NE cells and contributes to their versatility in generating various cell types within the nervous system [102].

In *O. masou*, the distribution patterns of GS+ cells largely correspond to the distribution of Nes+ cells [33], which indicates a high neurogenic activity and stemness of this zone. However, a Nes+- and BrdU-negative status in some of these cells suggests their dormant state [33,105].

In intact juvenile *O. keta*, GS+ type 1 cells have been found in various areas of the cerebellar body, in the superficial layers of the dorsal, lateral, and basal regions, where they were represented by single cells or formed hypercellular areas [33]. RG cells expressing GS were not found in *O. keta*, but patterns of migrating GS+ type 4 cells were quite common, which differed from the results of immunolabeling on *O. masou* [33]. In *O. masou*, single GS+ cells with RG morphology were identified, but the NE phenotype was also dominant. In the granular layer and granular eminence of *O. keta*, patterns of cell migration along guides were also revealed. The previously identified precursors of both types in *O. masou* probably indicate a certain relationship between embryonic-type NSCs and adult neuronal stem cells (aNSCs) in juveniles, while no GS+ RG cells were found in the *O. keta* cerebellum.

In *O. keta*, oval type 2 cells were observed in all neuroanatomical zones of the cerebellar body and parenchymal areas of the granular layer, which agrees with the data on *O. masou* [33]. In both species, these cells were characterized by a moderate or high intensity of GS labeling and were part of local networks providing neurochemical and homeostatic activity in the cerebellum of juveniles of both Pacific salmon species. Within the ganglionic layer of all neuroanatomical zones in the juvenile *O. keta* cerebellum, as in *O. masou*, large cells with a differentiated phenotype were found, but the intensity of the immunolabeling of such cells in intact *O. keta* was low (Table 2).

In adult humans and rodents, GS is expressed by subpopulations of oligodendrocytes [59]. Ultrastructural studies on the lizard midbrain showed that GS-positive cells may also be oligodendrocytes [106]. Furthermore, oligodendrocytes of different sizes and shapes were detected in the brain of the rainbow trout *O. mykiss* [107] using NADP-diaphorase enzyme histochemistry, and these proved to be morphologically similar to those of GS-positive cells of the cat shark mantle telencephalon. Differences in the sizes of the oligodendrocytes are related to the sizes of the axons they cover [108].

### 3.4. Expression of Glutamine Synthetase Post Injury

As a result of primary injury, the number of GS+ type 1 cells in the cerebellum of *O. keta* increased many times (Figure 5H). After a long primary post-injury period, juvenile *O. keta* showed a reorganization of neurogenic niches, in which the number of type 1 cells increased multifold. The size and frequency of occurrence of GS+ hypercellular areas in the molecular layer of the *O. keta* cerebellum increased significantly (Figure 5A,B), and multidirectional patterns of cell migration were observed in the granular layer (Figure 5D,F) and granular eminence (Figure 5G), as well as dense GS+ cell aggregations in the area of injury (Figure 5F). The results obtained showed an increase in the number of hypercellular foci, as well as vascularization patterns, while neuroepithelial structures exhibited complex histoarchitectonics and often contained apoptotic-type cellular debris surrounded by a hypercellular stroma. All these post-injury changes indicate significant variations in the cellular structure and a gradual increase in the population of GS+ cells as a result of a long primary post-injury period.

However, in contrast to acute traumatic injury in *O. masou* [33], there was no noticeable increase in the intensity of the GS immunolabeling in cells of various types in *O. keta* (Table 2). Thus, a long post-injury period, in contrast to an acute one, which is characterized by an increase in the immunolabeling intensity, in our opinion, is accompanied by plastic rearrangements in neurons of various types involved in the glutamate metabolism.

Previously, it was hypothesized that in case of acute injury to the *O. masou* cerebellum, type 3 cells, which regulate post-traumatic glutamate homeostasis in the cerebellum, are activated in order to utilize the toxic effects of glutamate and prevent increasing excitotoxicity accompanied by a reaction of secondary inflammation [13]. In *O. keta*, the number of GS+ cells in the cerebellum increases multifold during a long post-injury period, but the intensity of their labeling does not change significantly, which, in our opinion, is the result of plastic adaptation to a post-injury increase in the level of glutamate and its subsequent retraction during metabolism.

The intensity of the immunolabeling of type 1 and 2 NE cells also did not noticeably vary during the long-term post-injury period in *O. keta*, which differs from the data on an acute cerebellar injury in *O. masou*, where type 1 and 2 cells were characterized by an increased intensity of GS labeling [33]. A comparative analysis of the distributions of such cells in the DZ, LZ, and BZ showed significant increases in type 1 cells in the molecular layers of all areas of the *O. keta* cerebellum (Figure 6G). Consequently, during a long post-injury period in the *O. keta* cerebellum, neuroepithelial niches are reorganized, and larger hypercellular foci (reactive neurogenic niches, RNNs) are formed with enhanced neovascularization in all areas of the cerebellum. After injury, there is also an increase in GS+ type 2 cells (possibly corresponding to reactive mammalian astrocytes) and patterns of type 4 cell migrations in different areas of the cerebellum, which is consistent with data on juvenile *O. masou* [33]. However, in the long post-injury period, there is no increase in the intensity of the GS labeling in cells of various types, which differs from the acute period. Cells with the RG phenotype were not detected in *O. keta* in the chronic post-injury period, which disagrees with the data on *O. masou* [33].

In adult zebrafish, the brain and spinal cord can regenerate due to the presence of various types of neural stem and progenitor cells (NSPCs), including neuroepithelial cells (NECs) and rapidly and slowly proliferating radial glial cells (RGCs) [3,100,109,110]. However, the potential of different types of precursors in generating diverse clones of nerve cells and the internal mechanisms regulating neural regeneration in various brain regions after injury are still poorly understood. Research conducted on fish species has demonstrated the significance of these models in uncovering the mechanisms behind the functional regeneration of the CNS in vertebrates [11,111,112,113]. The presence of numerous neurogenic niches containing diverse combinations of NSPCs in the cerebellum of juvenile Pacific salmon (*O. keta*) supports previous findings in *O. masou* [33], suggesting further opportunities for studying the biology of NSPCs in the Pacific salmon brain.

In some earlier publications, a view of the biology and fate of aNSCs in the adult brain was presented, and aNSCs were assumed to retain a significant degree of multipotency, which made it possible to obtain different cell lines of the human brain after injury [113,114]. However, with the heterogeneous nature of the aNSC population in the spinal cord [114,115], in the midbrain tectum [114,116], and in the cerebellar neurogenic niches and the vagus nerve niches in the hindbrain [114,117], questions arise about the cell type that is at the top of the pedigree hierarchy and the degree of conservatism in various populations of aNSCs.

Studies on the regenerative potential and biology of aNSCs in various fish species [110,113,118,119,120] have shown that individual populations of aNSCs have various regenerative properties and different molecular controls. It remains to be clarified whether these properties are manifested due to specific reparative programs or due to a change in the aNSC microenvironment as a result of a TBI. The dissimilar patterns of NSPCs and the heterogenous mechanisms of activity of these cells are critical for the creation of correct post-traumatic programs associated with regeneration.

We suggest that one of the features of the juvenile *O. keta* cerebellum, as well as the *O. masou* cerebellum, is the ability to attract and mobilize different types of NSPCs, although GS+ RGCs were not detected in *O. keta* in the studied age period. In particular, the activation of various types of resting NSCs is an important property of ultrastructural and molecular rearrangement revealed both under constitutive conditions and after injury [102]. The increase in the production of new cells in the injured fish brain is likely regulated at both NSC and NCP levels. The rapid replacement of lost cells is connected with the elimination of progenitors needed to increase the final amplification pool [102]. The use of enhanced progenitor proliferation is typical and distinguishing feature for the rapid growth and enlargement of the cerebellum surface in the postembryonic development of fish [120,121,122,123], which largely agrees with the data on mammals [111].

Upon repeated injury to the cerebellum of juvenile *O. keta*, we detected a further increase in the number of GS+ cells which, however, was less pronounced than in the long-term period after the primary injury (Figure 6G). Nevertheless, a characteristic feature is the increase in the intensity of the GS immunolabeling in cells of all types, compared to the chronic period, which may be associated with the increased utilization of extracellular glutamate, which is typical of the acute post-injury phase. Thus, after a repeated acute traumatic injury, the amount of GS+ NSPCs continued to increase in the juvenile *O. keta* cerebellum, which indicates a lack of depletion in the pool of these cells, as well as after the repeated injury, when aromatase B was labeled.

In *D. rerio*, neuroepithelial and radial type NSPCs are significantly activated after damage to the cerebellum [124]. The activation of NE cells is suggested to lead to a rapid replenishment of the pool of granular cells that were lost as a result of unilateral ablation, while the activation of ventricular RG leads to a limited neurogenesis in *D. rerio* [102]. The results of our studies are consistent with the data obtained on *O. masou* [33] and *D. rerio* and indicate the predominance of NE type cells over RG cells during repeated injury in the acute period.

Cells of the ganglionic layer in the cerebellum LZ of juvenile *O. keta* intensely expressed GS after the repeated injury (Figure 6D). According to data on *D. rerio*, differentiated PC and EDC clones do not regenerate after a month post hatching [111]. Thus, the increases in GS activity in the EDCs and PCs of juvenile *O. keta* after the repeated injuries are associated with the intensification of glutamate metabolism and its reuptake from the intercellular space, whose volume increases multifold after acute traumatic injury. The emergence of migration patterns of type 1 cells and single GS+ RG cells corresponds to the data on acute traumatic injury in *O. masou*, and this obviously indicates a similar pattern of morphological and physiological changes after the repeated injury in *O. keta*.

### 3.5. Expression of Cystathionine-Beta Synthase in Adult Neurogenesis

It is generally accepted that CBS is the major hydrogen sulfide-producing enzyme in the vertebrate brain [125]. Polysulfide (H_2_S_n_)-associated signaling molecules such as hydrogen persulfide and trisulfide (H_2_S_2_ and H_2_S_3_) contribute to the maintenance of neurotransmission in neuronal networks, vascular tone, cytoprotection, inflammation, and oxygen sensitivity. H_2_S accelerates the initiation of hippocampal long-term potentiation (LTP) in the mammalian brain, which is a synaptic model for memory development, by enhancing the function of NMDA receptors [126]. CBS is expressed in the brain, and H_2_S activity in neurons stimulates Ca^2+^ flow between astrocytes and neurons to regulate the synaptic function [127].

In the intact juvenile *O. keta* cerebellum, weak or moderate CBS labeling was detected in cells of various types in the DMZ (Table 3), which disagrees with the data on the *O. masou* cerebellum, where moderate and high levels of immunolabeling intensity were detected in small type 1 and 2 cells in the DMZ [33]. In the dorsal and lateral regions of the *O. keta* cerebellum, patterns of the constitutive migration of CBS+ cells and moderately immunolabeled RG fibers were revealed. In the DZ and LZ, no large CBS+ hypercellular aggregations were found, in contrast to *O. masou*, in which hypercellular CBS+ aggregations of type 1 and 2 cells with superficial localizations were previously found in the DZ [33]. A morphotophographic analysis of the properties of these domains in *O. masou* showed a high degree of similarity with Nes+ and Vimentin+ and GS+ in CNNs [33]. In juvenile *O. keta*, constitutive hypercellular CBS+ aggregations were found only in the BZ, which suggests that H_2_S-producing cell populations of types 1 and 2 are involved in the formation of the intercellular environment and probably modulate the activity of CNNs containing NSPC markers.

As mammalian studies previously showed, CBS is localized specifically in astrocytes dispersed across six layers of the cortex, the dentate gyrus of the hippocampus, the layer of Purkinje, the *corpus callosum*, and the olfactory bulb, as well as in Bergmann’s glia of the cerebellum [128]. Subsequently, its colocalization with neuronal markers in the cerebral cortex has also been reported [129], as well as its expression in Purkinje cells and hippocampal neurons [130]. It is important to note that CBS in the mammalian brain is expressed in neuronal stem cells, where it appears to regulate their proliferation and differentiation [131,132].

In contrast to *O. masou*, which showed a more generalized distribution of intensely labeled CBS type 1 and 2 cells, the intense H_2_S-producing areas in intact juvenile *O. keta* were limited to the BZ, and only moderately weak basic CBS immunopositivity was recorded. Thus, in contrast to *O. masou* [33], the trout *O. mykiss* [133], and the cyprinid *C. carpio* [134], no high level of H_2_S production was found in most of the intact juvenile *O. keta* cerebellum.

### 3.6. Expression of Cystathionine-Beta Synthase Post Injury

As a result of long-term primary traumatic injury, the patterns of CBS immunolocalization in the *O. keta* cerebellum changed significantly. The most noticeable changes were associated with the emergence of hypercellular clusters in the molecular layer of the DZ and LZ containing intensely labeled type 1 and 2 cells. Significant histological rearrangements were also detected in the ganglionic and granular layers, where local complexes of type 1 cells in combination with RG and complexly organized dense hypercellular aggregations of CBS+ cells surrounded by a hypocellular stroma and possibly containing apoptotic fragments were visualized. Eventually, numerous patterns of the neovascularization and multidirectional migration of CBS+ cells were revealed in the juvenile *O. keta* cerebellum after the primary injury.

As was shown earlier, H_2_S is involved in the regulation of cell death signaling [135]. The death of neurons after injury is an important factor in the formation of neurological deficits [136,137]. In the *O. keta* cerebellum after primary injury, extensive cavities devoid of cells were found in the dorsal part of the granular layer (Figure 8E). It was reported that traumatic injury often induces the local apoptosis of neurons within a few hours [136].

According to Sarkar and co-authors, autophagy is a physiological process that maintains a balance between cell production and death [138]. While autophagy maintains the balance between the production of cellular components and the destruction of damaged organelles and other toxic cellular components under intact conditions in the juvenile *O. keta* cerebellum, the autophagic clearance is impaired as a result of injury, which correlates with neuronal death and agrees with data on mammals [139]. Thus, autophagy, as an attractive therapeutic target which, using new models such as, in particular, juvenile Pacific salmon, can be transformed into new therapeutic strategies to achieve better results in human TBI research [140].

In mammalian studies, H_2_S has been shown to regulate autophagy-dependent cell death after TBI [141,142,143]. In particular, it was found that 3-MST is mainly present in live neurons and may be involved in neuronal autophagy and in the pathophysiology of the brain after TBI [144]. As a result, brain cells become sensitive to ischemia [145].

We also associate the patterns of hypercellular aggregations producing H_2_S in the juvenile *O. keta* cerebellum with the cytoprotective effect of this gas transmitter. The increases in the size and number of H_2_S-producing hypercellular formations in the molecular and granular layers are consistent with the data on juvenile *O. masou* [33] and indicate the involvement of H_2_S in acute and chronic post-injury responses.

A study on long-term injury to the *O. keta* cerebellum showed a marked increase in the neovascularization in the dorsal part. It was previously reported that H_2_S dilates cerebral vessels after TBI by activating ATP-sensitive K-channels (KATP-channels) [146]. According to other data, specific molecular targets of H_2_S are cysteine 6 and 26 in the extracellular part of the rvSUR1 subunit of the K_ATP_ channel complex [147]. The opening of K_ATP_ channels inactivates the cell membrane polarization and voltage-gated calcium channels, which, in turn, leads to a decrease in Ca^2+^ content, eventually resulting in the relaxation and dilation of blood vessels.

During the repeated injury in the juvenile *O. keta* cerebellum, we observed a further increase in the number of CBS+ cells; however, the typical hypercellular clusters, found after the primary injury, were not detected after the repeated acute injury in *O. keta* [33]. The distribution patterns of intensely labeled CBS+ cells in the superficial regions of the molecular layer and the radial migration of CBS+ cells in the molecular and granular layers of juvenile *O. keta* resembled those in *O. masou* after a primary acute injury [13].

It is known that H_2_S mediates the intracellular calcium concentration [127,148,149]. After primary TBI, the disruption of the mitochondrial membrane potential (MMP) and the intracellular excess of reactive oxygen species (ROS) lead to accumulation of intracellular calcium, causing the calcium overload-induced neurotoxicity [150].

As our study has shown, the total number of CBS+ cells in the juvenile *O. keta* cerebellum, vice versa, increases significantly after injury, as in *O. masou*. In our opinion, these cells actively metabolize extracellular glutamate, reducing the intracellular production of ROS, thus eliminating the further development of oxidative stress. H_2_S is involved in many signaling pathways that attenuate the consequences of oxidative stress, particularly in the glutathione cycle, and in the activation of enzymes and transcription factors related to the redox balance [151]. Thus, one of the post-traumatic effects in the juvenile *O. keta* cerebellum is a further increase in the H_2_S production, which contributes to the maintenance of cerebrovascular homeostasis, including anti-apoptotic, anti-inflammatory, antioxidant, and pro-neurogenic effects that help reduce excitotoxic damage to neurons.

## 4. Material and Methods

### 4.1. Experimental Animals

The study was carried out on 60 juvenile individuals of the Pacific chum salmon *Oncorhynchus keta*, aged 1 year and 4 months, with a body length of 18–31 cm and a weight of 41–49 g. The animals were provided by the Ryazanovka experimental fish hatchery in 2021. Most of the fish used in this study were males. The juvenile chum salmon were kept in a tank with aerated fresh water at a temperature of 13–14 °C and fed once a day. The light/dark cycle was 14/10 h. The dissolved oxygen concentration in the water was 7–10 mg/dm^3^, which corresponds to normal saturation. All the experimental manipulations with animals were carried out in accordance with the rules regulated by the charter of the A.V. Zhirmunsky National Scientific Center of Marine Biology, Far Eastern Branch, Russian Academy of Sciences (NSCMB FEB RAS) and the Ethical Commission regulating the humane treatment of experimental animals (approval #1-110723 from Meeting No. 1 of the Commission on the biomedical ethics of NSCMB FEB RAS, 11 July 2023). The animals were divided into three groups. The animals in the control group (group 1) were intact (*n* = 20); the experimental groups consisted of fish exposed to a primary long-term traumatic cerebellar injury (group 2, *n* = 20) and a repeated acute traumatic injury (group 3, *n* = 20).

### 4.2. Experimental Design: Primary Long-Term Injury and Secondary Acute Injury

To study the characteristics of the cellular response as a result of long-term traumatic injury (group 2), the animals were subjected to a primary mechanical damage to the cerebellar region, followed by monitoring for 90 days post injury. The primary injury to the cerebellum was performed by puncturing the chum salmon’s skull according to the technique by Zupank and Ott [152]: a 1 mm deep wound was inflicted with a thin sterile needle in the dorsal region of the cerebellum body. The area of injury covered both the molecular and granular layers of the cerebellum body and did not affect other parts of the brain. Immediately after the mechanical damage, the animals were released into a tank with fresh water for recovery and further monitoring.

At 90 days post injury, the animals (group 2) were withdrawn from the experiment and processed for an IHC study. To study the characteristics of the cellular response after the repeated acute injury, the animals with the primary injury (group 3) were subjected to a repeated injury to the cerebellum region, followed by monitoring in a tank at 15 °C for 1 week post injury. After the second period, the animals (group 3) were withdrawn from the experiment and processed for the IHC study.

### 4.3. Preparation of Material for Immunohistochemical Studies

Anesthesia, prefixation. After the initial stage of injury (long-term primary injury, group 2) and after the repeated acute injury (group 3). The fish were anesthetized in a 0.1% tricaine methanesulfonate solution (MS222) (Sigma, St. Louis, MO, USA, Cat. no. WXBC9102V) for 10–15 min. After the anesthesia, the intracranial cavity of the immobilized animal was perfused with a 4% paraformaldehyde solution (PFA, BioChemica, Cambridge, MA, USA; Cat. No. A3813.1000; lot 31000997) in 0.1 M phosphate buffer (Tocris Bioscience, Minneapolis, MN, USA; Catalog No. 5564, Batch No. 5, pH 7.4). The animals were withdrawn from the experiment and euthanized via rapid decapitation. After the prefixation, the brain was removed from the cranial cavity and fixed for 24 h in a 4% paraformaldehyde solution in 0.1 M phosphate buffer. Then, they were kept in a 30% sucrose solution at 4 °C for two days (with seven changes of the solution). Serial frontal (50 μm thick) sections of the chum salmon brain were cut on a freezing microtome (Cryo-star HM 560 MV, Walldorf, Germany). Every third frontal section of the cerebellum was taken for the reaction.

### 4.4. Immunohistochemical Detection of Aromatase B, Glutamine Synthase, and Cystathionine-Beta-Synthase

To study the expressions of aromatase B, glutamine synthetase, and cystathionine-beta-synthase in the juvenile *O. keta* cerebellum after the primary long-term (90 days) and acute, repeated (1 week) traumatic injury to the cerebellum, immunoperoxidase labeling was used on frozen, free-floating brain sections. Before IHC, endogenous peroxidase activity and non-specific staining (background) were blocked via incubation with 1% hydrogen peroxide for 20 min at room temperature. To eliminate non-specific staining, the brain regions were incubated with non-immune horse serum.

After the incubation and washing with 0.1 M PBS, the frozen brain sections were incubated in situ with primary Goat Anti-Rabbit Anti-*cyp19a1b* antibody (Abcam, Cambridge, UK; Catalog No. ab106168), primary rabbit anti-CBS CBS polyclonal antibody (Abcam, Cambridge CB2 0AX, UK; catalog No. ab54883), and mouse anti-GS monoclonal antibodies (Abcam, Cambridge, UK; Catalog No. ab64613) diluted 1:300 at 4 °C for 48 h. The anti-rabbit streptavidin–biotin imaging system (HRP conjugated Anti-Rabbit IgG SABC Kit; Boster Biological Technology, Pleasanton, CA, USA; Catalog No. SA1022) and the avidin–biotin (ABC) complex (Vectastain Elite ABC kit; Vector Laboratories, San Francisco, CA, USA; Catalog No. PK-6100) were used to visualize the IHC labeling of polyclonal rabbit and monoclonal mouse antibodies, respectively. To identify the reaction products, a red substrate (VIP Substrate Kit, Cat. No. SK-4600, Vector Laboratories, Burlingame, CA, USA) was used in accordance with the manufacturer’s recommendations.

The cerebellum sections were placed on polylysine-coated glass slides (BioVitrum, St. Petersburg, Russia) and left to dry completely. To identify immunonegative cells, the cerebellum sections were additionally stained with a 0.1% methyl green solution (Bioenno Lifescience, Santa Ana, CA, USA, Cat. No. 003027). Color development was monitored under a microscope, washed in three changes of distilled water for 10 s, after which they were differentiated for 1–2 min in a 70% alcohol solution and then for 10 s in 96% ethanol. The sections were dehydrated according to the standard procedure: two xylene replacements, 15 min each. Then, they were placed under coverslips in a Bio-optica medium (Milano, Italy). The negative control method was used to assess the specificity of the immunohistochemical reaction (Figure 11A–C). The cerebellum sections, instead of primary antibodies, were incubated with a 1% solution of non-immune horse serum for 1 day and processed as sections with primary antibodies. In all control experiments, there was no immunopositive reaction.

### 4.5. Microscopy

For visualization and morphological and morphometric analyses of the cell body parameters (measurements of the greater and lesser diameters of soma), a research-grade motorized inverted microscope with an attachment for improved contrasting was used: Axiovert 200 m luminescence with an ApoTome fluorescent module and AxioCam MRM and AxioCam HRC digital cameras (Carl Zeiss, Jena, Germany). The material was analyzed using the AxioVision (Axiovert 200 M, Axiovision software version 4.8; Zeiss, Jena, Germany). Measurements were taken at 100×, 200×, and 400× magnifications and in several randomly selected fields of view for each region of interest. The number of labeled cells in the field of view was counted at the 200× magnification. Microphotographs of preparations were taken with an Axiovert 200 digital camera. The material was processed using the Axioimager program and in the Corel Photo-Paint 12 Graphic Editor.

### 4.6. Densitometry

The optical densities (ODs) of the IHC labeling products in neuron bodies and immunopositive granules were measured using the Axiovert 200 M microscope (Axiovert 200 M, Axiovision software version 4.8; Zeiss, Jena, Germany). The Wizard program was used to perform a standard estimation of the optical density for 5–7 sections by selecting 10–15 intensely/moderately labeled and immunonegative cells of the same type for analysis. Next, the average value of optical density for each cell type was subtracted from the maximum value of optical density for immunonegative cells (background), and thus, the actual value in relative units of optical density (UODs) was obtained.

### 4.7. Statistical Analysis

Morphometric data of IHC labeling were quantitatively processed using the Statistica version 12 and Microsoft Excel 2010 and STATA software packages (version 12, StataCorp LP., College Station, TX, USA). All data were presented as the mean ± standard deviation (M ± SD) and analyzed using the SPSS software application (version 12.0; SPSS Inc., Chicago, IL, USA). All group changes were compared using the Student–Newman–Keuls test or a one-way analysis/two-way analysis of variance (ANOVA, Chicago, IL, USA) with the Bonferroni correction. Values at *p* ≤ 0.01 and *p* ≤ 0.05 were considered statistically significant.

### 4.8. Stereological Method in the Study of the Quantitative Parameters of the Cerebellum

To obtain reliable quantitative characteristics of various regions of the juvenile *O. keta* cerebellum as a three-dimensional shape in space, we used the stereological method for calculating the data obtained through the microscopic analysis. For a reliable spatial reconstruction for the study of IHC, we used every three sections of the cerebellum. The stereological method allows, with reliability determined based on the objectives of the study and by varying the parameters of the study design, for the identification of the morphometric characteristics of the object under study on the material represented by a limited number of sections. In this case, it is proven that the systematic error (bias) was excluded, and the measurement error is controllable and directly depends on the sampling frequency: the more sections, the higher the accuracy. This is achieved through the use of an appropriate mathematical apparatus and compliance with sampling rules, in particular, systematic random sampling. When examining regions of the cerebellum, we selected the area of interest, after which we took into account the morphometric parameters of interest to us. After obtaining data from all the selected sections, we carried out calculations that allowed us to proceed to the description of the three-dimensional shape. In particular, data were obtained for such parameters as the number of immunopositive cells in the dorsal, lateral, and basal zones, in the granular eminence and in the granular layer, as well as for the distribution density of immunopositive cells in the constitutive and reactive neurogenic niches of all parts of the cerebellum.

## 5. Conclusions

The cerebellum is a generally accepted area of the brain for studying morphogenetic processes, since it has increased proliferative activity, both under normal conditions and after TBI [1]. The superficial location of the cerebellum facilitates visualization and direct and accurate access in case of injury. The architectonics and development of the cerebellum in teleost fish are well studied; the cerebellum is a conserved brain structure in vertebrates [153]. The cerebellum is functionally important for the integration of sensory perception and motor control [154,155,156]. Although the structure and development of the cerebellum has been most extensively analyzed in mammals, studies on fish have shown that the anatomy and development of the cerebellum are similar between mammalian species and the studied teleost fish [157].

Thus, despite the existing specific features of the neuroanatomical organization, the cerebellum of teleost fish has many features in common with the cerebellum of other vertebrates and is a good model system for studying the function and development of the cerebellum. The cerebellum integrates sensory input, which includes proprioception and information associated with motor commands, to induce fine motor control and modulate higher cognitive/emotional functions [158,159]. Elucidating the function and development of the cerebellum not only improves our understanding of the mechanisms of higher brain function, but may also suggest strategies for developing new therapies for diseases associated with the cerebellum.

Our study has made it possible to assess the dynamics of changes in the expressions of Aro B, GS, and CBS in various parts of the juvenile *O. keta* cerebellum. The different numbers of Aro+, GS+, and CBS+ cells are an indicator of their discreteness, which obviously indicates separate cellular expressions of these markers. The results of a two-way ANOVA test showed that in the cerebellum of intact chum salmon, the number of CBS+ in all zones exceeded the numbers of Aro+ and GS+ cells (*p* ≤ 0.05). The number of GS+ cells in intact animals was two to three times lower than that of CBS+ cells in all areas of the cerebellum. The number of Aro+ cells could be considered the most variable in intact *O. keta* (Figure 10).

As a result of the primary long-term TBI, the total numbers of Aro, GS, and CBS immunopositive cells increased significantly (Figure 12); however, the individual post-injury increases in each marker vary. In all the zones of the cerebellum, except the BZ, an approximately twofold increase in the number of CBS+ cells was recorded. The number of GS+ cells also increased approximately two- to three-fold in all the areas of the cerebellum (Figure 12). The variation in the number of Aro+ cells after the primary injury was the most pronounced and was characterized by specific dynamics for different areas of the cerebellum (Figure 12).

Upon the repeated acute TBI, the total numbers of Aro, GS, and CBS immunopositive cells continued to increase in all the cerebellum regions (Figure 13). However, the pattern of increasing dynamics was also strictly specific for each marker (Figure 13). In the DZ, LZ, granular layer, and granular eminence, the increases in Aro+ and CBS+ cells prevailed over the increase in the number of GS+ cells (Figure 13). In the BZ, the level of this prevalence was reduced (Figure 13).

Thus, the markers we discovered in the cerebellum of juvenile chum salmon likely act as neuroprotectors, since they have a protective effect by preventing excitotoxicity, inflammation, and oxidative damage; inhibiting apoptosis; promoting the survival of neurons; and regulating synaptogenesis. In response to the primary and repeated TBI to the cerebellum in chum salmon, numerous signaling pathways activated, which are potential candidates for the role of inducers of an “astrocyte-like” response in the cerebellum of juvenile *O. keta*, similar to reactions in the mammalian brain. We suggest that estradiol from Aro+ cells exerts a paracrine neuroprotective effect through the potential inhibition of inflammatory pathways. These results highlight the new importance of neuronal aromatization as a mechanism against the development of neuroinflammation. Our study has shown that the total numbers of CBS+ and GS+ cells in the juvenile cerebellum of *O. keta* increases significantly after primary and repeated TBI. In our opinion, these cells actively metabolize extracellular glutamate, reducing the intracellular production of reactive oxygen species and, thereby, eliminate the further development of oxidative stress. Thus, the post-traumatic decrease in extracellular glutamate in the juvenile cerebellum of *O. keta* and further increase in H_2_S production helps maintain cerebrovascular homeostasis, including anti-apoptotic, anti-inflammatory, antioxidant, and pro-neurogenic effects, which help reduce excitotoxic neuronal damage.

## Figures and Tables

**Figure 1 ijms-25-03299-f001:**
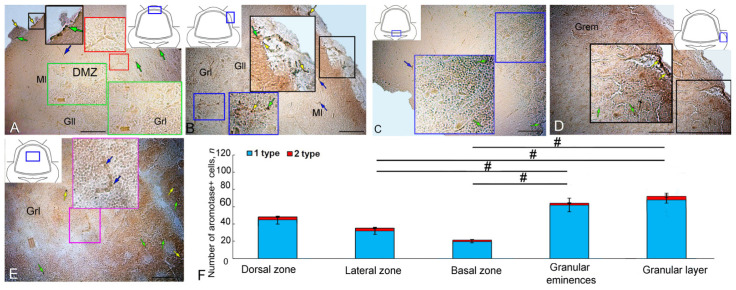
Representative images of the aromatase (Aro) distribution in the cerebellum of intact juvenile chum salmon, Oncorhynchus keta. (**A**) Dorsal zone: Aro+ elongated cells (blue arrows); Aro+ rounded type 1 cells (green arrows); Aro+ type 2 cells (yellow arrows); a cluster of Aro+ elongated cells (red inset); the dorsal matrix zone (DMZ) is highlighted in the green inset; superficial neurogenic niche containing Aro+ cells (black inset). (**B**) Lateral zone: Aro+ elongated cells (blue arrows); Aro+ rounded type 1 cells (green arrows); Aro+ type 2 cells (yellow arrows); blue inset indicates Аro+ cells in the granular layer; superficial neurogenic niche containing Aro+ cells (black inset). (**C**) Basal zone: Aro+ elongated cells (blue arrows); Aro+ rounded type 1 cells (green arrows); blue inset indicates Аro+ cells in the granular layer. (**D**) Granular eminence: the black insets show clusters of Aro+ rounded type 1 cells (green arrows) and Aro+ cells type 2 (yellow arrows). (**E**) Granular layer: Aro+ rounded type 1 cells (green arrows); Aro+ type 2 cells (yellow arrows); Aro+ elongated cells (blue arrows); pink inset indicates Aro+ cells in the granular layer. (**F**) Aro+ cell ratios in different areas of the cerebellum in intact brain; # *p* ≤ 0.05, significant differences between the zones of the intact brain (*n* = 5 in each group); one-way analysis of variance (ANOVA), F = 2.358. ML—molecular layer, Gll—ganglionic layer, Grl—granular layer, Grem—granular eminence. IHC labeling of aromatase. The blue inset on the pictogram indicates the respective zone in the micrograph. Scale: (**A**–**E**) 100 µm.

**Figure 2 ijms-25-03299-f002:**
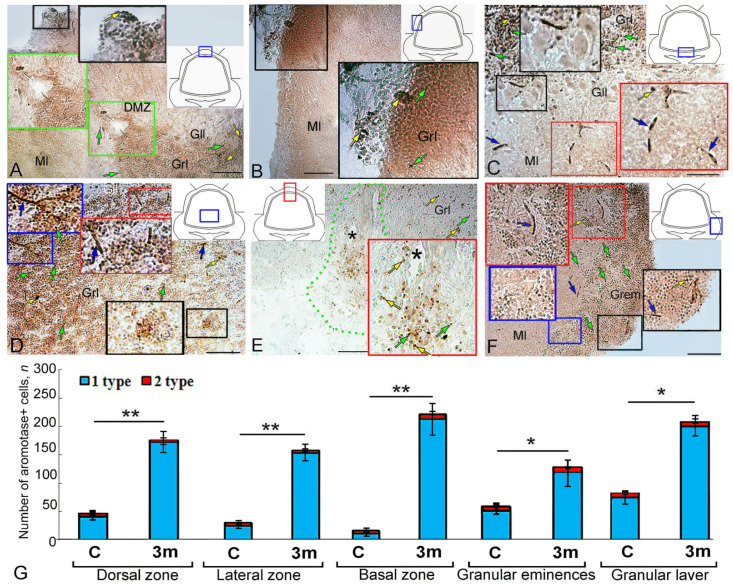
Representative images of the aromatase (Aro) distribution in the cerebellum of juvenile chum salmon, *O. keta,* at 3 months post brain injury. (**A**) Dorsal zone: Aro+ rounded type 1 cells (green arrows); Aro+ type 2 cells (yellow arrows); the dorsal matrix zone (DMZ) is highlighted in the green insets; superficial neurogenic niche (black inset). (**B**) Lateral zone: Aro+ rounded type 1 cells (green arrows); Aro+ type 2 cells (yellow arrows); superficial neurogenic niche (black inset). (**C**) Basal zone: Aro+ rounded type 1 cells (green arrows); Aro+ type 2 cells (yellow arrows); Aro+ elongated cells (blue arrows); red insets show elongated Aro+ cells, black insets show Aro-negative projection neurons. (**D**) Granular layer: + rounded type 1 cells (green arrows); Aro+ type 2 cells (yellow arrows); Aro+ elongated cells (blue arrows); red and blue insets show clusters of Aro+ elongated cells; the black insets outline a cluster of Aro+ cells. (**E**) Area of damage, highlighted with a green dotted line. * indicates a zone of injury. The red insets show clusters of Aro+ rounded type 1 cells (green arrows) and Aro+ cells type 2 (yellow arrows). (**F**) Granular eminence: Aro+ rounded type 1 cells (green arrows); Aro+ type 2 cells (yellow arrows); Aro+ elongated cells (blue arrows); black insets show Aro+ cells near the superficial layer; red inset, Aro + elongated cells; blue inset, individual Aro+ cells. (**G**) Quantitative proportions of Aro+ cells in control and 3 months post traumatic injury. (*n* = 5 in each group; * *p* ≤ 0.05 and ** *p* ≤ 0.01—significant differences vs. control groups). Student–Newman–Keuls test. ML—molecular layer, Gll—ganglionic layer, Grl—granular layer, Grem—granular eminence. C—intact brain, 3m—three months after injury. IHC labeling of aromatase. The blue insets in the pictograms show the respective zones in the micrographs. The red inset on the pictogram indicates the zone of injury. Scale: (**A**–**F**) 100 µm.

**Figure 3 ijms-25-03299-f003:**
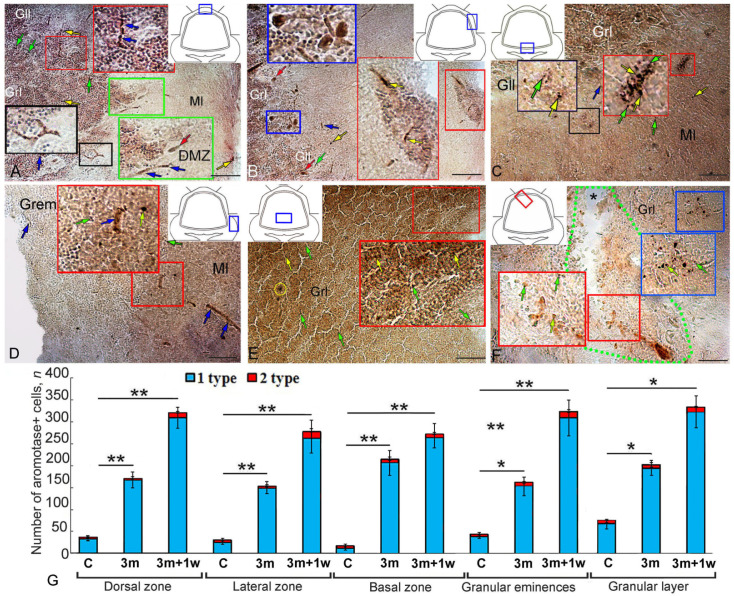
Representative images of the aromatase (Aro) distribution in the cerebellum of juvenile chum salmon *Oncorhynchus O. keta* at 3 months and 1 week post brain injury. (**A**) Dorsal zone: Aro+ elongated cells (blue arrows); Aro+ rounded type 1 cells (green arrows); Aro+ type 2 cells (yellow arrows); red arrows show Aro+ projection neurons; the dorsal matrix zone (DMZ) is highlighted the in green insets; Aro+ elongated cells are shown in the black insets; neurogenic niche (red inset). (**B**) Lateral zone: Aro+ elongated cells (blue arrows); Aro+ rounded type 1 cells (green arrows); Aro+ type 2 cells (yellow arrows); red arrows show Aro+ projection neurons; blue insets show Aro+ projection neurons; neurogenic niche (red inset). (**C**) Basal zone: Aro+ elongated cells (blue arrows); Aro+ rounded type 1 cells (green arrows); Aro+ type 2 cells (yellow arrows); red and black insets indicate clusters of Aro+ cells. (**D**) Granular eminence: Aro+ elongated cells (blue arrows); Aro+ rounded type 1 cells (green arrows); Aro+ type 2 cells (yellow arrows); the red insets show Aro+ cells. (**E**) Granular layer: Aro+ rounded type 1 cells (green arrows); Aro+ type 2 cells (yellow arrows). Red inset indicate clusters of Aro+ cells; the yellow solid oval shows a cluster of Aro+ cells. (**F**) Area of damage, highlighted in green dotted line. * indicates a zone of injury: Aro+ rounded type 1 cells (green arrows); Aro+ type 2 cells (yellow arrows); red and blue insets show clusters of Aro+ cells. (**G**) Quantitative proportions of Aro+ cells in control and after traumatic injury (*n* = 5 in each group; * *p* ≤ 0.05 and ** *p* ≤ 0.01 are significant vs. control groups). Student–Newman–Keuls test. ML—molecular layer, Gll—ganglionic layer, Grl—granular layer, Grem—granular eminence. C—intact brain, 3m—three months post injury, 3m + 1w—three months and 1 week after injury. IHC labeling of aromatase. The blue insets in the pictograms show the respective zones in the micrographs. The red inset in the pictogram shows the zone of injury. Scale: (**A**–**F**) 100 µm.

**Figure 4 ijms-25-03299-f004:**
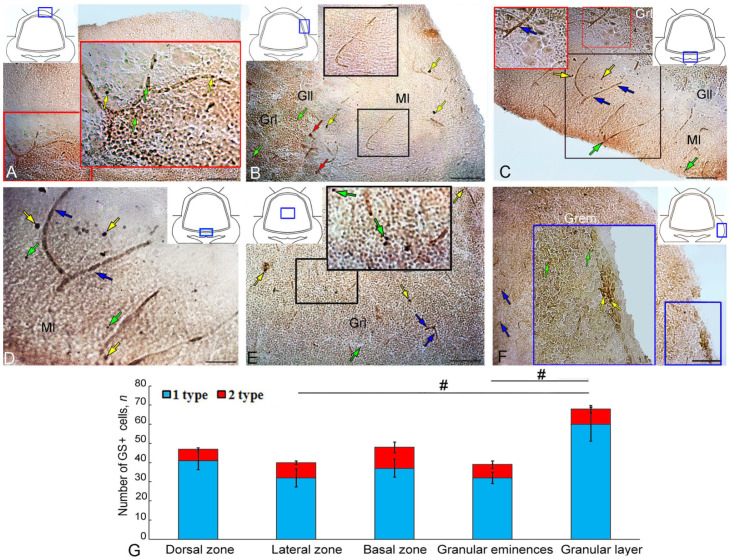
Representative images of the distribution of glutamine synthetase (GS) in the intact cerebellum of juvenile chum salmon, *Oncorhynchus O. keta*. (**A**) Dorsal zone: GS+ rounded of type 1 cells (green arrows); GS+ of type 2 cells (yellow arrows); a cluster of GS+ cells (red inset). (**B**) Lateral zone: GS+ rounded of type 1 cells (green arrows); GS+ of type 2 cells (yellow arrows); black insets show GS+ cells; red arrows show GS+ projection neurons. (**C**) Basal zone: GS+ elongated cells (blue arrows); GS+ rounded type 1 cells (green arrows); GS+ type 2 cells (yellow arrows); a cluster of GS+ cells (red inset); the black inset shows a magnified fragment in (**C**). (**D**) Enlarged fragment of the basal zone: GS+ elongated cells (blue arrows); Aro+ rounded type 1 cells (green arrows); GS+ type 2 cells (yellow arrows). (**E**) Granular layer: GS+ elongated cells (blue arrows); GS+ rounded type 1 cells (green arrows); GS+ type 2 cells (yellow arrows); GS+ cells are highlighted in the black insets. (**F**) Granular eminence: GS+ elongated cells (blue arrows); GS+ rounded type 1 cells (green arrows); GS+ type 2 cells (yellow arrows); the blue insets show GS+ cells in the superficial layer. (**G**) The proportions of GS+ cells in different areas of the cerebellum in the intact brain; # *p* ≤ 0.05, significant differences between zone of the intact brain (*n* = 5 in each group); one–way analysis of variance (ANOVA), F = 2.45. ML—molecular layer, Gll—ganglionic layer, Grl—granular layer, Grem—granular eminence. IHC labeling glutamine synthetase. The blue insets on the pictograms indicate the corresponding zones in the micrographs. Scale: (**A**–**C**,**E**,**F**) 100 µm; (**D**) 50 µm.

**Figure 5 ijms-25-03299-f005:**
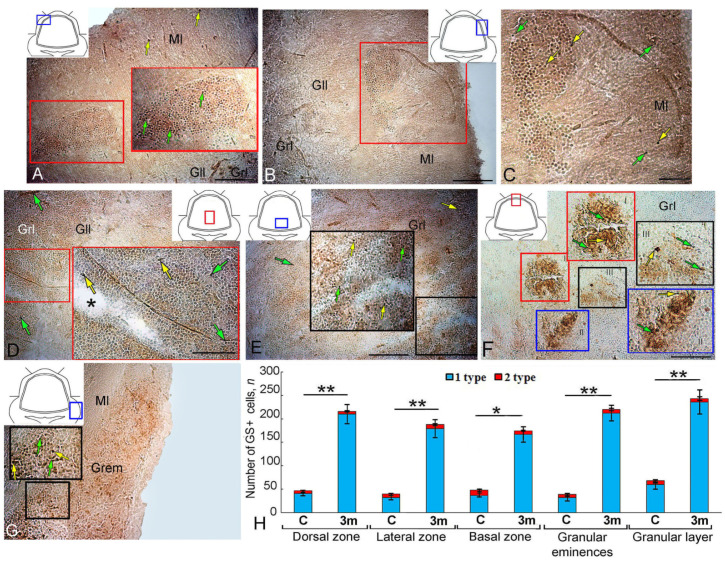
Representative images of the distribution of glutamine synthetase (GS) in the cerebellum of juvenile chum salmon, *Oncorhynchus O. keta,* at 3 months post brain injury. (**A**) Dorsal zone: GS+ rounded type 1 cells (green arrows); GS+ type 2 cells (yellow arrows); red insets show GS+ cells in the reactive neurogenic niche (**B**) Lateral zone: the red inset shows a magnified fragment in (**C**) (reactive neurogenic niche). (**C**) Enlarged fragment of the reactive neurogenic niche: GS+ rounded type 1 cells (green arrows); GS+ type 2 cells (yellow arrows) (**D**) Area of damage in granular layer: GS+ rounded type 1 cells (green arrows); GS+ type 2 cells (yellow arrows); * indicates a zone of injury. The red inset shows clusters of GS+ elongated cells. (**E**) Granular layer: GS+ rounded type 1 cells (green arrows); GS+ type 2 cells (yellow arrows); the black insets show GS+ cells of types 1 and 2. (**F**) Area of damage: GS+ rounded type 1 cells (green arrows); GS+ type 2 cells (yellow arrows); I, II, and III show aggregations of GS+ cells. (**G**) Granular eminence: GS+ rounded type 1 cells (green arrows); GS+ type 2 cells (yellow arrows); the black insets show GS+ cells of types 1 and 2. (**H**) Quantitative ratios of GS+ cells in control and at 3 months after traumatic injury. (*n* = 5 in each group; * *p* ≤ 0.05 and ** *p* ≤ 0.01 are significant differences vs. control groups). Student–Newman–Keuls test. ML—molecular layer, Gll—ganglionic layer, Grl—granular layer, Grem—granular eminence. C—intact brain, 3m—three months after injury. IHC labeling glutamine synthetase. The blue insets in the pictograms show the respective zones in the micrographs. The red insets in the pictograms indicate the zones of injury. Scale: (**A**,**B**,**D**–**G**) 100 µm; (**C**) 50 µm.

**Figure 6 ijms-25-03299-f006:**
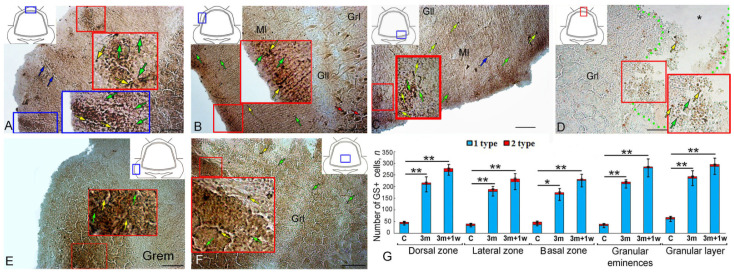
Representative images of the distribution of glutamine synthetase (GS) in the cerebellum of juvenile chum salmon, *Oncorhynchus O. keta,* at 3 months and 1 week post brain injury. (**A**) Dorsal zone: GS+ elongated cells (blue arrows); GS+ rounded cells of type 1 (green arrows); GS+ rounded type 2 cells (yellow arrows); reactive neurogenic niches containing GS+ cells are shown in blue and red insets. (**B**) Lateral zone; GS+ rounded cells of type 1 (green arrows); GS+ rounded type 2 cells (yellow arrows); red arrows show GS+ projection neurons; red insets show cluster of GS+ cells. (**C**) Basal zone: GS+ elongated cells (blue arrows); GS+ rounded cells of type 1 (green arrows); GS+ rounded type 2 cells (yellow arrows); red insets show clusters of GS+ cells. (**D**) Area of damage, highlighted with a green dotted line; * indicates a zone of injury. GS+ rounded cells of type 1 (green arrows); GS+ rounded type 2 cells (yellow arrows); red insets show clusters of GS+ cells. (**E**) Granular eminence: GS+ rounded type 1 cells (green arrows); GS+ type 2 cells (yellow arrows); the red inset highlights the aggregation of GS+ cells. (**F**) Granular layer: GS+ rounded cells of type 1 (green arrows); GS+ rounded type 2 cells (yellow arrows); the red inset highlights the aggregation of GS+ cells. (**G**) Quantitative proportions of GS+ cells in the control and after traumatic injury. (*n* = 5 in each group; * *p* ≤ 0.05 and ** *p* ≤ 0.01 are significant differences vs. control groups). Student–Newman–Keuls test. ML—molecular layer, Gll—ganglionic layer, Grl—granular layer, Grem—granular eminence. C—intact brain, 3m—three months post injury, 3m + 1w—three months and 1 week post injury. IHC labeling glutamine synthetase. The blue insets on the pictograms indicate the corresponding zones in the micrographs. The red inset on the pictogram indicates the zone of injury. Scale: (**A**–**F**) 100 µm.

**Figure 7 ijms-25-03299-f007:**
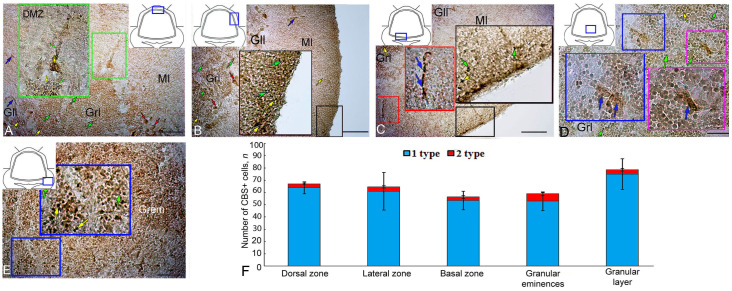
Representative images of the distribution of cystathionine β–synthase (CBS) in the cerebellum of intact juvenile chum salmon, *Oncorhynchus O. keta*. (**A**) The dorsal matrix zone is shown in the green insets; CBS+ elongated cells (blue arrows); CBS+ rounded 1 cells type (green arrows); CBS+ type 2 cells (yellow arrows); red arrows show GS+ projection neurons. (**B**) Lateral zone: CBS+ elongated cells (blue arrows); CBS+ rounded 1 cells type (green arrows); CBS+ type 2 cells (yellow arrows); red arrows show GS+ projection neurons; CBS+ surface layer cells are shown in the black insets. (**C**) Basal zone:CBS+ elongated cells (blue arrows); CBS+ rounded 1 cells type (green arrows); CBS+ type 2 cells (yellow arrows); CBS+ surface layer cells are shown in black inset; red insets show chains of CBS+ cells. (**D**) Granular layer: CBS+ elongated cells (blue arrows); CBS+ rounded 1 cells type (green arrows); CBS+ type 2 cells (yellow arrows). Blue and pink insets show clusters of CBS+ cells. (**E**) Granular eminence: CBS+ rounded 1 cells type (green arrows); CBS+ type 2 cells (yellow arrows). Blue inset shows CBS+ cells. (**F**) The proportion of CBS+ cells in different zones of the cerebellum in the intact brain (*n* = 5 in each group); one-way analysis of variance (ANOVA), F = 1.366. ML—molecular layer, Gll—ganglionic layer, Grl—granular layer, Grem—granular eminence. IHC labeling of cystathionine β–synthase. The blue insets in the pictograms show the respective zones in the micrographs. Scale: (**A**–**E**) 100 µm.

**Figure 8 ijms-25-03299-f008:**
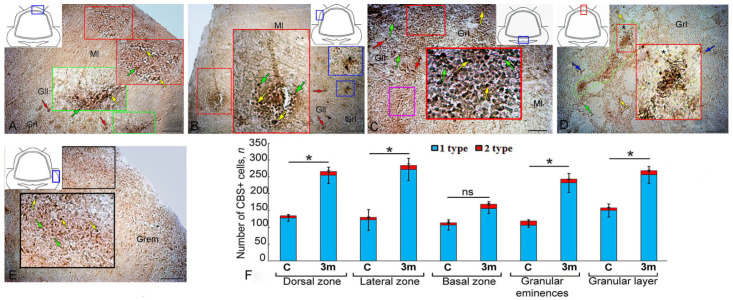
Representative images of the distribution of cystathionine–β–synthase (CBS) in the cerebellum of juvenile chum salmon, *Oncorhynchus O. keta*, at 3 months after brain injury. (**A**) Dorsal matrix zone (shown in green insets): CBS+ rounded type 1 cells (green arrows); CBS+ type 2 cells (yellow arrows); red arrows show CBS + projection neurons; red insets show reactive neurogenic niches containing CBS+ type 1 and type 2 cells. (**B**) Lateral zone: CBS+ rounded type 1 cells (green arrows); CBS+ type 2 cells (yellow arrows); red arrows show CBS + projection neurons; red insets show reactive neurogenic niches containing CBS+ type 1 and type 2 cells; blue insets show clusters of CBS+ cells in the granular layer. (**C**) Basal zone: CBS+ rounded type 1 cells (green arrows); CBS+ type 2 cells (yellow arrows); red arrows show CBS + projection neurons; the pink inset shows elongated CBS+ cells; red insets show CBS+ cells. (**D**) Area of damage in the granular layer: highlighted with a green dotted line, * indicates a zone of injury. CBS+ rounded type 1 cells (green arrows); CBS+ type 2 cells (yellow arrows); red insets show the clustering of CBS+ cells; CBS+ elongated cells (blue arrows). (**E**) Granular eminence: CBS+ rounded type 1 cells (green arrows); CBS+ type 2 cells (yellow arrows); clusters of CBS+ cells are highlighted by black insets. (**F**) Quantitative ratios of CBS+ cells in the control and 3 months after traumatic injury (*n* = 5 in each group; ns—no significant differences, * *p* ≤ 0.05 and ** *p* ≤ 0.01 are significant differences vs. control groups). Student–Newman–Keuls test. ML—molecular layer, Gll—ganglionic layer, Grl—granular layer, Grem—granular eminence. IHC labeling of cystathionine β–synthase. C—intact brain, 3m—three months post injury. The blue insets in the pictograms show the respective zones in the micrographs. The red inset in the pictogram shows the zone of injury. Scale: (**A**–**E**) 100 µm.

**Figure 9 ijms-25-03299-f009:**
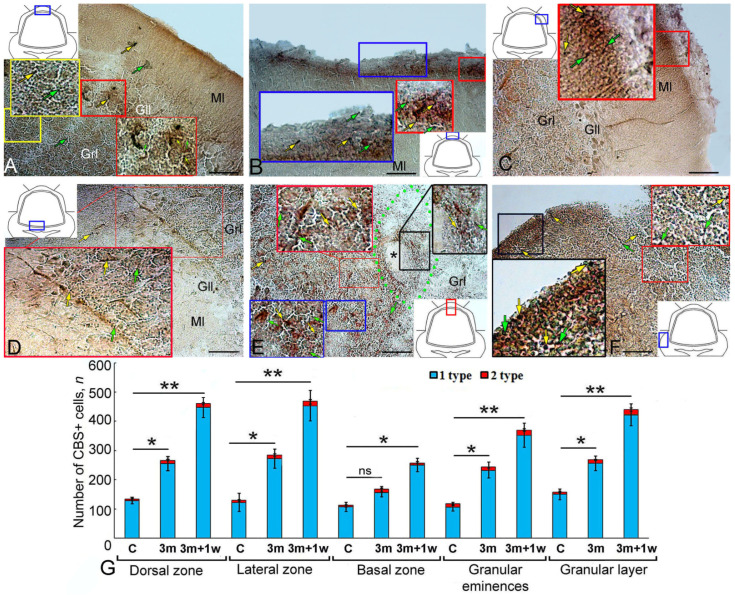
Representative images of the distribution of cystathionine–β–synthase (CBS) in the cerebellum of juvenile chum salmon, *Oncorhynchus O. keta,* at 3 months and 1 week post brain injury. (**A**) Dorsal zone: CBS+ rounded type 1 cells (green arrows); CBS+ rounded type 2 cells (yellow arrows); the aggregation of CBS+ cells is highlighted in the red insets; elongated CBS+ cells are shown in the yellow insets. (**B**) Dorsal zone: CBS+ rounded type 1 cells (green arrows); CBS+ rounded type 2 cells (yellow arrows); red and blue insets show accumulations of CBS+ cells. (**C**) Lateral zone: CBS+ rounded type 1 cells (green arrows); CBS+ rounded type 2 cells (yellow arrows); the aggregation of CBS+ cells is highlighted in the red insets. (**D**) Basal zone: CBS+ rounded type 1 cells (green arrows); CBS+ rounded type 2 cells (yellow arrows); the red insets shows cluster of CBS+ cells. (**E**) Area of damage: highlighted with a green dotted line; * indicates a zone of injury. CBS+ rounded type 1 cells (green arrows); CBS+ rounded type 2 cells (yellow arrows); red, blue, and black insets highlight the accumulations of CBS+ cells. (**F**) Granular eminence: CBS+ rounded type 1 cells (green arrows); CBS+ rounded type 2 cells (yellow arrows); CBS+ cells are highlighted in red and black insets. (**G**) Quantitative ratios of CBS+ cells in control and after traumatic injury (*n* = 5 in each group; ns—no significant differences, * *p* ≤ 0.05 and ** *p* ≤ 0.01 are significant differences vs. control groups). Student–Newman–Keuls test. ML—molecular layer, Gll—ganglionic layer, Grl—granular layer, Grem—granular eminence. IHC labeling of cystathionine β–synthase. C—intact brain, 3m—three months post injury, 3m + 1w—three months and 1 week post injury. The blue insets in the pictograms show the respective zones in the micrographs. The red inset in the pictogram shows the zone of injury. Scale: (**A**–**F**) 100 µm.

**Figure 10 ijms-25-03299-f010:**
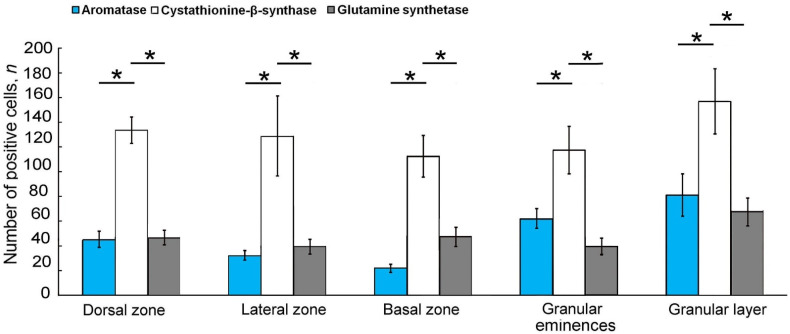
Total distributions of Aro+, GS+ and CBS+ cells in the cerebellum of intact juvenile chum salmon, O. keta, (*n* = 5 in each group; * *p* ≤ 0.05). Two–way analysis of variance (ANOVA). The cerebellum zones are plotted on the *X*-axis.

**Figure 11 ijms-25-03299-f011:**
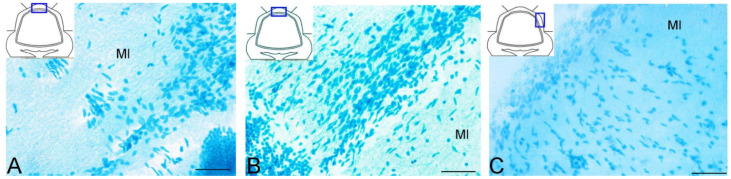
Negative controls (representative sections) in the cerebellum of juvenile chum salmon, *Oncorhynchus keta*. (**A**) IHC labeling of aromatase in the dorsal cerebellum. (**B**) IHC labeling of GC in the dorsal cerebellum. (**C**) IHC labeling of CBS in the lateral zone of the cerebellum. The blue inset in the pictogram shows the respective zone in the micrograph. Scale bar: (**A**–**C**) 50 µm.

**Figure 12 ijms-25-03299-f012:**
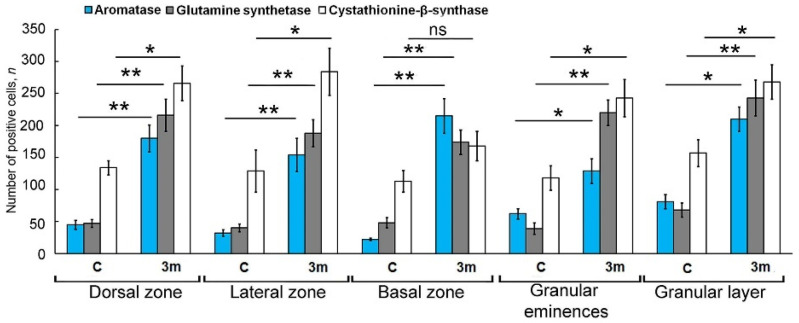
Total distributions of Aro+, GS+ and CBS+ cells in the intact cerebellum of juvenile chum salmon *Oncorhynchus keta* and at three months post injury. (*n* = 5 in each group; ns—no significant differences, * *p* ≤ 0.05 and ** *p* ≤ 0.01 are significant differences vs. control groups). Student–Newman–Keuls test. The cerebellum zones are plotted on the *X*-axis.

**Figure 13 ijms-25-03299-f013:**
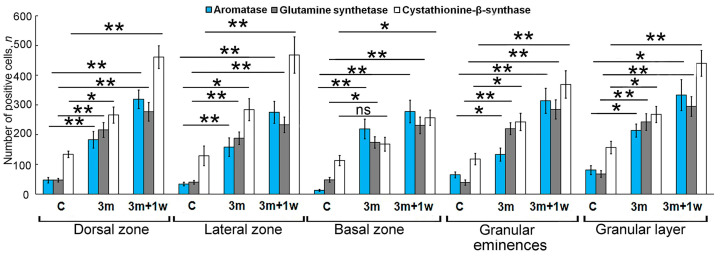
Total distributions of Aro+, GS+ and CBS+ cells in the intact cerebellum of juvenile chum salmon, *Oncorhynchus keta,* at 3 months post injury and 3 months and 1 week post injury. (*n* = 5 in each group; ns—no significant differences, * *p* ≤ 0.05 and ** *p* ≤ 0.01 are significant differences vs. control groups). Student–Newman–Keuls test. The cerebellum zones are plotted on the *X*-axis.

**Table 1 ijms-25-03299-t001:** Morphometric and densitometric characteristics of aromatase labeled cells (M ± SD) in the intact and injured cerebellum (3 months and 3 months + 1 week) of juvenile chum salmon, *Oncorhynchus keta*.

	Intact Animals			Long-Term TBI		Long-Term TBI + Acute TBI	
	Aromatase						
Brain Areas	Type of cell	Cell Size, µm	Optical Density *, UOD	Cell Size, µm	Optical Density *, UOD	Cell Size, µm	Optical Density *, UOD
**Dorsal zone**	1 type, round	4.4 ± 0.3/4.1 ± 0.2	121.2 ± 11.1/168.5 ± 13.4	4.6 ± 0.3/3.1 ± 0.3	136.1 ± 13.4/198.5 ± 19.7	3.9 ± 0.4/3.2 ± 0.4	141.3 ± 10.2/208.5 ± 17.7
	2 type, oval	6.1 ± 0.3/5.1 ± 0.4	133.4 ± 11.7/172.4 ± 16.7	6.2 ± 0.5/4.6 ± 0.3	146.3 ± 11.4/184.2 ± 19.1	5.7 ± 0.5/5.2 ± 0.3	155.4 ± 15.2/202.3 ± 21.3
	3 type, differentiated	–	–	–	–	21.4 ± 1.4/11.6 ± 1.7	147.4 ± 11.2
	4 type, elongated	11.5 ± 0.7/4.8 ± 0.4	71.8 ± 9.7/121.2 ± 11.2	–	–	14.4 ± 0.9/4.6 ± 0.3	153.4 ± 10.1/182.9 ± 16.6
**Lateral zone**	1 type, round	4 ± 0.3/4.2 ± 0.5	139.2 ± 11.3/178.3 ± 10.8	4.3 ± 0.2/3.7 ± 0.3	158.5 ± 12.4/189.7 ± 20.4	4.5 ± 0.2/4.1 ± 0.2	199.6 ± 19.5
	2 type, oval	5.9 ± 0.3/4.6 ± 0.5	126.4 ± 15.2/188.6 ± 11.2	5.3 ± 0. 2/4.7 ± 0.3	133.4 ± 9.2/199.4 ± 17.2	5.5 ± 0.4/4.7 ± 0.4	139.3 ± 15.1/198.5 ± 17.7
	3 type, differentiated	–	–	–	–	28.6 ± 2.5/17.1 ± 1.4	125.3 ± 12.1/205.9 ± 21.2
	4 type, elongated	12.4 ± 1.1/5.2 ± 0.3	121.9 ± 12.4	–	–	13.2 ± 0.5/5.2 ± 0.4	133.5 ± 11.2/200.5 ± 19.3
**Basal zone**	1 type, round	4.2 ± 0.4/4.5 ± 0.4	137.3 ± 10/168.2 ± 12.6	4.1 ± 0.3/3.1 ± 0.1	136.6 ± 11.3/191.3 ± 17.9	4.8 ± 0.6/3.2 ± 0.2	144.3 ± 12.4/199.5 ± 17.9
	2 type, oval	5.7 ± 0.5/4.3 ± 0.3	126.6 ± 8.2/178.7 ± 10.7	5.5 ± 0.3/4.9 ± 0.3	119.3 ± 9.1/199.2 ± 18.1	5.6 ± 0.5/5.1 ± 0.6	132.4 ± 11.1/197.8 ± 15.7
	3 type, differentiated	–	–	–	–	29.6 ± 2.6/17.3 ± 1.2	122.7 ± 8.9/178.5 ± 20.3
	4 type, elongated	11.6 ± 0.9/4.9 ± 0.5	82.4 ± 10.1/126.2 ± 11.6	12.1 ± 0.6/4.7 ± 0.6	199.5 ± 19.1	13.5 ± 0.6/5.5 ± 0.6	141.5 ± 10/188.5 ± 17.7
**Granular layer**	1 type, round	4.5 ± 0.4/4.3 ± 0.3	131.1 ± 9.3/177.3 ± 13.4	4.4 ± 0.4/3.5 ± 0.3	141.6 ± 10.2/191.5 ± 16.8	4.3 ± 0.2/3.6 ± 0.3	137.8 ± 10.2/191.3 ± 18.2
	2 type, oval	5.9 ± 0.3/4.8 ± 0.6	139.6 ± 11.3	6.2 ± 0.5/4.7 ± 0.3	178.7 ± 17.2	6 ± 0.4/4.6 ± 0.5	199.1 ± 19.7
	4 type, elongated	–	–	12.5 ± 1.3/4.4 ± 0.5	181.2 ± 15.3	14.7 ± 0.9/4.6 ± 0.5	141.4 ± 12.3/192.3 ± 21.4
**Granular eminence**	1 type, round	3.9 ± 0.3/3.7 ± 0.4	143.3 ± 11.4	4.1 ± 0.2/4.1 ± 0.1	128.4 ± 8.2/181 ± 19.1	4.8 ± 0.6/4 ± 0.5	149.2 ± 12.1/199.9 ± 17.4
	2 type, oval	4.9 ± 0.3/4.3 ± 0.2	122.4 ± 12.5/188.2 ± 8.1	6.4 ± 0. 3/5 ± 0.3	146.7 ± 11.1/182.5 ± 17.3	5.8 ± 0.8/4.8 ± 0.6	141.4 ± 10/204.97 ± 22.4
	4 type, elongated	–	–	12.5 ± 1.3/4.4 ± 0.5	192.3 ± 16.3	11.9 ± 1.4/4.8 ± 0.6	189.3 ± 17.1

* Optical density (OD) in cells was classified according to the following scale: high (210–160 UOD), moderate (160–110 UOD), and weak (less than 110 UOD). The large and small diameters of the cell body are shown with a slash (/). Cells moderately and intensely labeled are shown with a slash (/). Long-term TBI—3 months after injury, long-term TBI + acute TBI—3 months and 1 week after injury.

**Table 2 ijms-25-03299-t002:** Morphometric and densitometric characteristics of GS labeled cells (M ± SD) in intact cerebellum and during long-term TBI and long-term TBI + acute TBI in juvenile chum salmon, *Oncorhynchus keta*.

	Intact Animals			Long-Term TBI		Long-Term TBI + Acute TBI	
	Glutamine Synthetase						
Brain Areas	Type of Cell	Cell Size, µm	Optical Density *, UOD	Cell Size, µm	Optical Density *, UOD	Cell Size, µm	Optical Density *, UOD
**Dorsal zone**	1 type, round 2 type, oval 4 type, elongated	3.7 ± 0.3/3.6 ± 0.3 5.8 ± 0.4/4.8 ± 0.3 – 9.1 ± 0.2/4.6 ± 0.2	103.2 ± 10.4/167.8 ± 16.3 123.2 ± 10.9 – 103 ± 10.2	3.9 ± 0.4/3.2 ± 0.4 5.7 ± 0.5/5.2 ± 0.1 – –	116.7 ± 11.3/161.7 ± 16.8 177.7 ± 18.4 – –	4.6 ± 0.3/3.5 ± 0.3 6.2 ± 0.5/4.6 ± 0.3 – 11.3 ± 0.7/5.6 ± 0.8	177.7 ± 17.8 158 ± 16.4 – 122.8 ± 12.3/177.7 ± 18.4
**Lateral zone**	1 type, round 2 type, oval 3 type, differentiated 4 type, elongated	4.1 ± 0.4/3.8 ± 0.5 5.7 ± 0.4/4.6 ± 0.2 23.5 ± 1.2/14.5 ± 0.3 11.3 ± 0.4/4.4 ± 0.7	126.4 ± 9.3/133.6 ± 15.2 117.5 ± 9.8 122.2 ± 12.1 67.1 ± 7.4	4.5 ± 0.2/4.1 ± 0.2 5.5 ± 0.4/4.7 ± 0.4 – –	153.2 ± 15.2 123.4 ± 12.2/173.9 ± 19.3 – –	4.3 ± 0.2/3.7 ± 0.3 5.3 ± 0.2/4.7 ± 0.3 33.1 ± 4.1/22.2 ± 1 11 ± 0.5/5.1 ± 0.5	167.6 ± 18.1 169.7 ± 13.4 129.2 ± 11.5/171.4 ± 14.8 133.7 ± 16.2/166.1 ± 12.1
**Basal zone**	1 type, round 2 type, oval 4 type, elongated	3.9 ± 0.2/3.7 ± 0.4 5.4 ± 0.3/4.7 ± 0.5 – 12.3 ± 0.6/4.8 ± 0.4	119.3 ± 12.4/157.3 ± 18.1 113.7 ± 10.2 – 123.2 ± 15.1	4.8 ± 0.6/3.2 ± 0.2 5.6 ± 0.5/5.1 ± 0.6 – –	113.2 ± 10.2/163.5 ± 19.8 176.8 ± 12.6 – –	4.1 ± 0.3/3.3 ± 0.4 5.5 ± 0.3/4.9 ± 0.3 – 11.4 ± 0.3/5 ± 0.4	179.1 ± 15.1 179.7 ± 12.4 – 132.8 ± 11.6/161.7 ± 12.5
**Granular layer**	1 type, round 2 type, oval 4 type, elongated	3.4 ± 0.2/3.2 ± 0.4 5.3 ± 0.3/4.1 ± 0.3 11.3 ± 0.3/5.2 ± 0.4	116.6 ± 9.9/148.6 ± 13.1 116.1 ± 11.8 110.9 ± 10.5	4.3 ± 0.2/3.6 ± 0.3 6 ± 0.4/4.6 ± 0.5 –	122.1 ± 11.4/173.1 ± 18.5 177.5 ± 14.2 –	4.4 ± 0.4/3.5 ± 0.3 6.2 ± 0.5/4.7 ± 0.3 11.9 ± 0.9/5.7 ± 0.5	179.6 ± 15.1 159.4 ± 12.3 129.9 ± 13.2
**Granular eminence**	1 type, round 2 type, oval 4 type, elongated	3.5 ± 0.4/3.4 ± 0.4 5.5 ± 0.4/4.4 ± 0.4 11.6 ± 0.2/5 ± 0.5	111.3 ± 11.2/153.3 ± 14.6 122.4 ± 11.6/134.7 ± 13.2 116.3 ± 11.6	4.8 ± 0.6/4 ± 0.5 5.8 ± 0.8/4.8 ± 0.6 12.3 ± 0.8/5.8 ± 0.6	163.1 ± 18.4 153.7 ± 15.2 129.2 ± 12.5/163.7 ± 17.2	4.1 ± 0.2/4.1 ± 0.1 6.4 ± 0.3/5 ± 0.3 11.9 ± 0.9/5.9 ± 0.7	179.4 ± 11.4 169.7 ± 10 111.5 ± 9.4/171.1 ± 11.1

***** Optical density (OD) in cells was classified according to the following scale: high (180–140 UOD), moderate (140–90 UOD), and weak (less than 90 UOD). * The large and small diameters of the cell body are shown with a slash (/). Cells moderately and intensely labeled are shown with a slash (/). Long-term TBI—3 months after injury, long-term TBI + acute TBI—3 months and 1 week after injury.

**Table 3 ijms-25-03299-t003:** Morphometric and densitometric characteristics of cystathionine β-synthase labeled cells (M ± SD) in the intact and long-term TBI (3 months) and long-term TBI + acute TBI (3 months + 1 week) of juvenile keta Salmon, *Oncorhynchus keta*.

	Intact Animals			Long-Term TBI		Long-Term TBI + Acute TBI	
	Cystathionine β-Synthase						
Brain Areas	Type of Cell	Cell Size, µm	Optical Density *, UOD	Cell Size, µm	Optical Density *, UOD	Cell Size, µm	Optical Density *, UOD
**Dorsal zone**	1 type, round	4.4 ± 0.5/3.3 ± 0.5	94.4 ± 10.7/101.4 ± 13	4.3 ± 0.3/3.7 ± 0.3	131 ± 5.6/182.4 ± 77.9	4.1± 0.3/3.9 ± 0.3	177.5 ± 18.1
	2 type, oval	5.9 ± 0.3/5.1 ± 0.3	91.3 ± 11.3/115.5 ± 11.3	6.1 ± 0.6/5.6 ± 0.5	152.3 ± 9.2	6 ± 0.4/5 ± 0.5	183.4 ± 17.7
	3 type, differentiated	18.9 ±1.5/15.6 ± 1.2	86.6 ± 7.3/114.2 ± 9.2	18.5 ±1.2/15.2 ± 1.5	96.4 ± 9.3/118.2 ± 10.2	19.3 ±1.1/17.1 ± 1.3	74.7 ± 11.2/101.7 ± 12.1
	4 type, elongated	10.2 ± 1.1/5.1± 0.3	91.5 ± 9.3/105.7 ± 9.6	10.2 ± 1.1/5.1± 0.3	124.5 ± 10.2/175.5 ± 11.9	–	–
**Lateral zone**	1 type, round	4.5 ± 0.4/3.4 ± 0.4	81.3 ± 9.4/123.9 ± 7.1	3.7 ± 0.3/3.4 ± 0.5	135.4 ± 4.6/167.5 ± 12.1	4.1± 0.3/3.5± 0.3	163.4 ± 9.2
	2 type, oval	5.5 ± 0.4/4.9 ± 0.3	86.2 ± 13.1/127.3 ± 9.2	5.7 ± 0.5/4.8 ± 0.3	183.4 ± 17.1	5.2± 0.5/4± 0.3	174.5 ± 6.9
	3 type, differentiated	19.1 ±2.1/15.1 ± 1.3	84.8 ± 9.2/133.5 ± 8.1	20.1 ±0.9/15.4 ± 1.2	89.8 ± 8.1/133.5 ± 18.5	20.1 ±0.9/15.4 ± 1.2	109.2 ± 10.3
	4 type, elongated	11.2 ± 1/5.3± 0.4	97.4 ± 8.3/115.9 ± 11.2	10.1 ± 1/5.2± 0.4	144.2 ± 9.4/164 ± 10	–	–
**Basal zone**	1 type, round	4.4 ± 0.3/3.6 ± 0.4	86.8 ± 5.9/123.9 ± 7.1/167.2 ± 12.3	4 ± 0.5/3.3 ±0.4	108.9 ± 11.2/157 ± 17.3	4. 4 ± 0.3/3.4 ± 0.2	188.2 ± 9.2
	2 type, oval	6.2 ± 0.3/4.7 ± 0.4	91.3 ± 9.9/133 ± 12.4/158.2 ± 14.6	6 ± 0.5/5.3 ± 0.2	111.2 ± 7.1/173.2 ± 18.1	5.7 ± 0.3/5.3 ± 0.2	173 ± 11.7
	3 type, differentiated	22.1 ±2.1/14.3 ± 1.1	77.7 ± 9.4/133.9 ± 11.1	19.4 ±1.1/15.2 ± 1.5	87.4 ± 11.5/127.6 ± 13.4	17.7 ± 1.3/16.2 ± 1.3	178.5 ± 15.6
	4 type, elongated	10.4 ± 1.3/5.3± 0.4	84.5 ± 13.6/116.9 ± 7.2	10.2 ± 1.1/5.1± 0.3	133.2 ± 11.2	11.2 ± 1/4.8± 0.4	173.6 ± 17.1
**Granular layer**	1 type, round	4.4 ± 0.5/3.9 ± 0.4	79.3 ± 8.7/143.6 ± 11.4	4.4 ± 0.2/3.4 ± 0.5	125.2 ± 11.4/163.2 ± 15.5	4.1 ±0.2/3.6 ±0.1	188.5 ± 12.4
	2 type, oval	6.7 ± 0.4/5.1± 0.6	89.8 ± 9.1/123.1 ± 12.5	5.9 ± 0.5/4.2 ± 0.5	139.7 ± 11.5/188.2 ± 18.8	6.1 ± 0. 3/5.1 ± 0.3	179.3 ± 9.6
	4 type, elongated	10 ± 0.9/5 ± 0.4	90 ± 10.7/133.5 ± 16.3	12.2 ± 1/4.9± 0.4	123.1 ± 12.5	12.1± 1.2/5.2 ± 0.5	181.3 ± 14.4
**Granular eminence**	1 type, round	4.3 ± 0.5/3.9 ± 0.4	123.6 ± 13.2	4.4 ± 0.3/3.3 ± 0.4	122.2 ± 11.3/163.8 ± 15.3	4.4 ± 0.2/3.4 ±0.1	183.7 ± 19.2
	2 type, oval	6.7 ± 0.4/4.9± 0.6	154.5 ± 17.1	5.9 ± 0.6/4 ± 0.2	126 ±10.2/173.4 ± 17.9	5.5 ± 0. 3/4.4 ± 0.3	179.5 ± 18.1
	4 type, elongated	–	–	12.2 ± 1/4.9± 0.4	137.1 ± 12.5	–	–

***** Optical density (OD) in cells was classified according to the following scale: high (190–150 UOD), moderate (150–100 UOD), and weak (less than 100 UOD). * The large and small diameters of the cell body are shown with a slash (/). Cells moderately and intensely labeled are shown with a slash (/). Long-term TBI—three months post injury, long-term TBI + acute TBI—three months and 1 week post injury.

## Data Availability

Data is contained within the article.

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
