# Peer review of "Post-Traumatic Expressions of Aromatase B, Glutamine Synthetase, and Cystathionine-Beta-Synthase in the Cerebellum of Juvenile Chum Salmon, Oncorhynchus keta"

_ijms, 2024, doi:10.3390/ijms25063299_

Round 1

Reviewer 1 Report (New Reviewer)

Comments and Suggestions for Authors

The manuscript “Comparative Analysis Post-Traumatic Dynamics of Expression of Aromatase B, Glutamine Synthetase, and Cystathionine-beta-Synthase in the Cerebellum of Juvenile Chum Salmon, Oncorhynchus keta” by Pushchina is a research article which examined the expression of romatase B (AroB), glutamine synthetase (GS), and cystathionine-beta-synthase (CBS) during a long-term primary traumatic injury and after a repeated acute traumatic injury to the cerebellum of Oncorhynchus keta juveniles. After the primary injury in the cerebellum of O. keta juveniles, the number of AroB+, GS+, CBS+ cells increased in the ganglionic and granular layers and the granular eminence. In summary, upon repeated injury, the number of neuronal stem/progenitor cells (NSPCs) in the juvenile O. keta cerebellum continued to increase without a noticeable decrease in the regenerative potential. In general, this article is critical in this field and contains essential results. However, I have several comments before this manuscript is accepted for publication.

1. The abstract is too long. The abstract should be less than 200 words. The authors should concisely summarize the results. The important point in this manuscript is to describe the expression changes of of Aromatase B, Glutamine Synthetase, and Cystathionine-beta-Synthase.

2. Also, the introduction is too long, and the critical rationale in this manuscript is not clear. Please rewrite the introduction. In this manuscript, the authors mainly examined the expression of Aromatase B, Glutamine Synthetase, and Cystathionine-beta-Synthase. Therefore, the Introduction should be focused on these three synthtase. The description of H2S is too long. Please shorten the description about H2S.

3. In bar graphs, all the data plots should be displayed if possible. The readers can obtain more information from these data.

4. In this manuscript, the statistical analysis was performed using one-way and two-way ANOVA. The F values should be added.

5. In the materials and methods, it is described that “most of the fish used in this work were males”. Have the authors checked the exact number of male fish?

6. The authors should make clear what is the difference in the neurogenesis between teleost fish and mammalian species.

Author Response

The manuscript “Comparative Analysis Post-Traumatic Dynamics of Expression of Aromatase B, Glutamine Synthetase, and Cystathionine-beta-Synthase in the Cerebellum of Juvenile Chum Salmon, Oncorhynchus keta” by Pushchina is a research article which examined the expression of romatase B (AroB), glutamine synthetase (GS), and cystathionine-beta-synthase (CBS) during a long-term primary traumatic injury and after a repeated acute traumatic injury to the cerebellum of Oncorhynchus keta juveniles. After the primary injury in the cerebellum of O. keta juveniles, the number of AroB+, GS+, CBS+ cells increased in the ganglionic and granular layers and the granular eminence. In summary, upon repeated injury, the number of neuronal stem/progenitor cells (NSPCs) in the juvenile O. keta cerebellum continued to increase without a noticeable decrease in the regenerative potential. In general, this article is critical in this field and contains essential results. However, I have several comments before this manuscript is accepted for publication.

  1. The abstract is too long. The abstract should be less than 200 words. The authors should concisely summarize the results. The important point in this manuscript is to describe the expression changes of of Aromatase B, Glutamine Synthetase, and Cystathionine-beta-Synthase.

Thank you for your comments, we have included the recommended abbreviation in the abstract, however, to comply with the recommendation, we have reduced the length to 270 words (more reduction will lead to the loss of important data).

  1. Also, the introduction is too long, and the critical rationale in this manuscript is not clear. Please rewrite the introduction. In this manuscript, the authors mainly examined the expression of Aromatase B, Glutamine Synthetase, and Cystathionine-beta-Synthase. Therefore, the Introduction should be focused on these three synthtase. The description of H2S is too long. Please shorten the description about H2S.

Thank you for this recommendation. In accordance with the recommendations made, the description of H2S has been shortened in the Introduction section. At the same time, critical rationale is added, including the multidimensionality of traumatic injury and potential therapeutic targets that this research may address.

  1. In bar graphs, all the data plots should be displayed if possible. The readers can obtain more information from these data.

We presented the graphs in Figures 10, 11, 12, which reflect the data we obtained. In particular, Figure 10 shows the comparative distribution of Aro+, GS+, SBS+ cells in various anatomical zones of the intact brain. (Total distribution of Аro+, GS+, CBS+ cells in the cerebellum of juvenile chum salmon O. keta in the intact brain). Next, in Figure 11, we visually compare the distribution of Aro+, GS+, CBS+ cells normally and after 3 months (Total distribution of Aro+, GS+, CBS+ cells in the cerebellum of juvenile chum salmon Oncorhynchus keta in the intact brain and three months after injury). And Figure 12 displays the final comparison of normal, chronic injury and acute injury when labeling Aro+, GS+ and CBS+ cells (Total distribution of Aro+, GS+, CBS+ cells in the cerebellum of juvenile chum salmon Oncorhynchus keta in the intact brain, 3 months after injury and 3 months and 1 week after injury). We reflected all the designations of markers and anatomical zones in the transcript of the captions to the figures. We would be grateful to the reviewer if he could suggest in more detail how best to present our data in Your opinion.

  1. In this manuscript, the statistical analysis was performed using one-way and two-way ANOVA. The F values should be added.

F values added for Fig. 1, 4 and 7.

  1. In the materials and methods, it is described that “most of the fish used in this work were males”. Have the authors checked the exact number of male fish?

Thanks for your question, 52 of the fish were males, 8 of the fish were unsure of their sex.

  1. The authors should make clear what is the difference in the neurogenesis between teleost fish and mammalian species.Thank you for this comment; the Introduction section provides additional support for differences in post-traumatic neurogenesis in fish and mammals.

Reviewer 2 Report (New Reviewer)

Comments and Suggestions for Authors

Comments to Authors

In this article entitled ‘Post-Traumatic Expression of Aromatase B, Glutamine Synthetase, and Cystathionine-beta-Synthase in the Cerebellum of Juvenile Chum Salmon, Oncorhynchus keta’

Pushchina et al., demonstrated the role of Aromatase B, Glutamine Synthetase, and Cystathionine-beta-Synthase in the Cerebellum of Juvenile Chum Salmon post trauma. The concept behind study is an excellent idea which was completely diluted with the poor quality of presentation.

All section of this manuscript including Abstract, introduction, discussion and conclusion are too long without any reason and these sections section should be shortened.

Use of abbreviation is not correct throughout the manuscript.

While results obtained are good but not presented well.

Immunohistochemical data presented in most figures are not of good quality because of poor preservation of tissue.

This study narrative should be focused on cerebellum and post trauma.

Smooth reading of this work is  interrupted due to unnecessary content and authors are encouraged to rewrite this MSS. 

Comments on the Quality of English Language

Poorly written manuscript

Author Response

Comments to Authors

In this article entitled ‘Post-Traumatic Expression of Aromatase B, Glutamine Synthetase, and Cystathionine-beta-Synthase in the Cerebellum of Juvenile Chum Salmon, Oncorhynchus keta’

Pushchina et al., demonstrated the role of Aromatase B, Glutamine Synthetase, and Cystathionine-beta-Synthase in the Cerebellum of Juvenile Chum Salmon post trauma. The concept behind study is an excellent idea which was completely diluted with the poor quality of presentation.

All section of this manuscript including Abstract, introduction, discussion and conclusion are too long without any reason and these sections section should be shortened.

Abstract and Introduction have been shortened in accordance with the recommendations. As recommended by another reviewer, the Introduction section has been updated to provide critical evidence supporting the need for these studies.

Use of abbreviation is not correct throughout the manuscript.

Why the use of abbreviations in the manuscript is unacceptable - it is incorrect. In many similar works, incl. Other authors widely use abbreviations to schematize and simplify the understanding of notations in the text.

While results obtained are good but not presented well.

This comment is not clear. A rating of “not good enough” is not a recommendation that specifies what exactly needs to be improved in a given work on its own merits.

Immunohistochemical data presented in most figures are not of good quality because of poor preservation of tissue.

In this study, tissue preservation was ensured by performing a histological fixation protocol immediately after the euthanasia procedure.

After the initial stage of injury (long-term primary injury, group 2) and after the repeated acute injury (group 3). The fish were anesthetized in a 0.1% tricaine methanesulfonate solution (MS222) (Sigma, St. Louis, MO, USA, Cat. no. WXBC9102V) for 10–15 min. After the anesthesia, the intracranial cavity of the immobilized animal was perfused with a 4% paraformaldehyde solution (PFA, BioChemica, Cambridge, MA, USA; Cat. No. A3813.1000; lot 31000997) in 0.1 M phosphate buffer (Tocris Bioscience, Minneapolis, MN, USA; Catalog No. 5564, Batch No. 5, pH 7.4). The animals were withdrawn from the experiment and euthanized by rapid decapitation. After the prefixation, the brain was removed from the cranial cavity and fixed for 24 h in a 4% paraformaldehyde in 0.1 M phosphate buffer. Then they were kept in a 30% sucrose solution at 4°C for two days (with a sevenfold change of the solution). Serial frontal (50 μm thick) sections of the chum salmon brain were cut on a freezing microtome (Cryo-star HM 560 MV, Walldorf, Germany). Every third frontal section of the cerebellum was taken for the reaction.

This study narrative should be focused on cerebellum and post trauma.

In accordance with the previously recommendations of other reviewers, this work involves consideration of:

1) comparative aspects of the reparative process in the brain of mammals and humans, as well as fish;

2) Critical justification for the study of the regenerative process in the cerebellum and the participation of GS, CBS, AroB in it;

3) Direct presentation of the data from this study on juvenile chum salmon, taking into account the endemicity of this species, as well as data on fetalization of the cerebellum

4) Various aspects of TBI, in particular, consideration of the prospects for using the results obtained on juvenile salmon in subsequent clinical practice;

Smooth reading of this work is  interrupted due to unnecessary content and authors are encouraged to rewrite this MSS. 

Significant changes have been made to the article in accordance with the recommendations of the reviewers, in particular, redundant sections have been shortened. The majority of reviewers (3 reviewers) considered the article sound and well-planned; moreover, the reviewers' recommendations included desires to expand the discussion by presenting considerations about the possible potential for use in clinical practice, comparisons of data with mammals, etc. All of these recommendations do not support this recommendation.

Reviewer 3 Report (New Reviewer)

Comments and Suggestions for Authors

The Manuscript deal with an interesting tool and is well structured, however the English has to be revised to make the manuscript more clear and other minor reviosions are needed.

- Report the abbreviation the first time and not the second (chek lines 19-20 and 46 and 53-54; lines 81-82; line 99)

- line 113-114 the authors refer to the Aromatase B? Why they use the abbreviation AroB only here? Were is the reference about is role in the TBI?

- I think there is a missing in this sentence "The cyp19a1 genes are regulated by brain-specific promoters, such as promoter [13], which are responsible for initiating gene expression specifically in the brain [21]"

- lines 137-139 and 154-156 are repetitive

- there is a mistake at line 263 "Apo+ cells"

- the negative control and figure 13 are not really clear

Comments on the Quality of English Language

The English needs to be revised to make clear the text (i.e. lines 97-98; 110-112;

These are only some exemples.

Grammar and spelling errors are also present.

Author Response

The Manuscript deal with an interesting tool and is well structured, however the English has to be revised to make the manuscript more clear and other minor reviosions are needed.

- Report the abbreviation the first time and not the second (chek lines 19-20 and 46 and 53-54; lines 81-82; line 99)

In accordance with the recommendations, the necessary changes related to abbreviations were made to the Introduction.

- line 113-114 the authors refer to the Aromatase B? Why they use the abbreviation AroB only here? Were is the reference about is role in the TBI?

In accordance with the recommendations, the necessary changes related to abbreviations were made to the Introduction. There was no previous mention of the involvement of AroB in TBI.

- I think there is a missing in this sentence "The cyp19a1 genes are regulated by brain-specific promoters, such as promoter [13], which are responsible for initiating gene expression specifically in the brain [21]"

As recommended, this section of the introduction has been shortened and replaced with additional critical support for this manuscript.

- lines 137-139 and 154-156 are repetitive

Thanks for the comment, corrected

- there is a mistake at line 263 "Apo+ cells"

Thanks for the comment, corrected

- the negative control and figure 13 are not really clear

Figure 13 shows representative versions of negative controls for AroB, GS, and CBS immunolabeling. In all cases, primary (specific antibodies) were absent. In the Materials and Methods section, the corresponding methodological descriptions for these sections are highlighted. If necessary to provide microphotographs of intact and all types of experimental immunolabelings of AroB, GS and CBS, we are ready to provide this.

Reviewer 4 Report (New Reviewer)

Comments and Suggestions for Authors

In the present experimental study, the Authors aimed to compare patterns of the expression of aromatase B, glutamine synthetase, and cystathionine-beta-synthase in the cerebellum of intact O. keta juveniles. To identify the dynamics that determine the involvement of AroB, GS, and CBS in cellular mechanisms during regeneration, they assessed the expression of these molecular markers during a long-term primary traumatic injury and a repeated acute traumatic injury to the juvenile O. keta cerebellum.

Overall, I found the study timely, original, well conducted and scientifically sound. However, I have some minor comments aimed at improving the very high quality of the paper, and these are outlined below:

1.     In the introduction, a brief note on the fact that traumatic injuries might be multidimensional which include several subtypes with different neurobiological underpinnings and several comorbidities, should be added with appropriate references.

2.     Translating into “real world” clinical practice and medicine, what possible clinical shreds of evidence might arise from the present study and what the Researchers do suggest improving clinical practice? Please add a brief paragraph on possible suggestions in terms of integrative care. I suggest expanding a bit and make deeper and clear the "conclusions" section.

Comments on the Quality of English Language

I suggest a check of the english language with the help of a native speaker

Author Response

In the present experimental study, the Authors aimed to compare patterns of the expression of aromatase B, glutamine synthetase, and cystathionine-beta-synthase in the cerebellum of intact O. keta juveniles. To identify the dynamics that determine the involvement of AroB, GS, and CBS in cellular mechanisms during regeneration, they assessed the expression of these molecular markers during a long-term primary traumatic injury and a repeated acute traumatic injury to the juvenile O. keta cerebellum.

Overall, I found the study timely, original, well conducted and scientifically sound. However, I have some minor comments aimed at improving the very high quality of the paper, and these are outlined below:

  1. In the introduction, a brief note on the fact that traumatic injuries might be multidimensional which include several subtypes with different neurobiological underpinnings and several comorbidities, should be added with appropriate references.

 Thank you for this comment. In accordance with the recommendation, relevant data supported by references from modern literature has been added to the Introduction.

  1. Translating into “real world” clinical practice and medicine, what possible clinical shreds of evidence might arise from the present study and what the Researchers do suggest improving clinical practice? Please add a brief paragraph on possible suggestions in terms of integrative care. I suggest expanding a bit and make deeper and clear the "conclusions" section.

 Thank you for this comment. Our proposed model of post-traumatic regeneration of juvenile Pacific chum salmon᾿s cerebellum suggests a high regenerative response associated with increased expression of AroB, GS and CBS during cerebellar injury, the potential of which does not further decrease with repeated TBI. Considering that the majority of immunopositive cells in the cerebellum expressing ApoB, GS and CBS have a neuroepithelial phenotype (type 1) and their number increases as a result of primary and secondary injury, we can conclude that as a result of TBI in the cerebellum of juvenile chum salmon there is an increase in the number of neurons. This is due to the absence of RG expressing these markers in the cerebellum of juvenile chum salmon, which corresponds to data on zebrafish (117) and differs from the results of studies on Apteronotus (121). NECs, as is known, predominate in the embryonic period of development in mammals (13-15) and fish (16, 17), divide by symmetrical mitosis and expand the neuronal progenitor pool (1,9,10). The resulting neuronal progenitors subsequently give rise to the neurons. Thus, in juvenile chum salmon with primary and secondary injuries to the cerebellum, the embryonic scenario of post-traumatic neurogenesis is preserved (i), in zones of primary proliferation and foci of secondary neurogenesis in the parenchyma, the number of cells expressing ApoB, GS and CBS increases, the action of which is aimed at neuroprotection, increasing the survival of neurons formed after TBI.

Round 2

Reviewer 1 Report (New Reviewer)

Comments and Suggestions for Authors

The authors addressed all my concerns.

Reviewer 2 Report (New Reviewer)

Comments and Suggestions for Authors

Authors response is not satisfactory and sufficient enough.

Comments on the Quality of English Language

All comments are same as before.

This manuscript is a resubmission of an earlier submission. The following is a list of the peer review reports and author responses from that submission.

Round 1

Reviewer 1 Report

Comments and Suggestions for Authors

The manuscript entitled “Comparative Analysis and Post-Traumatic Dynamics of Expression of Aromatase B, Glutamine Synthetase, and Cystathionine-beta-Synthase in the Cerebellum of Juvenile Chum Salmon, Oncorhynchus keta”, by Pushchina et al., is aimed at the immunocytochemical localization of the above-mentioned three molecules in the brain of a teleost fish.

Unfortunately, the results are based on images that are not at all convincing, since in most photographs the staining with respect to background is barely visible and do not seem specific.

The text is extremely long and wordy, yet, most of all, it cannot rely on the results showed in the Figures (see comment above on the quality of images/staining.

For these reasons, I think that this manuscript cannot be published as it is.

Author Response

This work presents data on IHC labeling of Aromatase B, GS, and CBS in the cerebellum of juvenile Pacific chum salmon. IHC labeling was performed according to previously published protocols ( Pushchina et al., 2021 ; Pushchina et al., 2022 ). The work also provides data from a negative analysis of immunostaining for all markers. Below we provide the necessary explanations for all types of immunolabeling and methods for their analysis and evaluation. We associate the low intensity of IHC labeling of AroB in cells 3 and 4 in intact juvenile chum salmon with their immature status of the studied juveniles and the fact that mainly males were used in the experiment. The cyp19a1b gene encoding AroB is expressed in the brain, but its expression pattern and regulation may vary depending on the fish species, sex and developmental stage (Diotel, N.; Le Page, Y.; Mouriec, K.; Tong, S. K.; Pellegrini, E.; Vaillant, C.; Anglade, I.; Brion, F.; Pakdel, F.; Chung, B. C.; et al. Aromatase in the brain of teleost fish: expression, regulation and putative functions. Front. Neuroendocrinol. 2010, 31, 172-192. DOI: 10.1016/j.yfrne.2010.01.003). It is known that an increase in aromatase B activity during the breeding season in fish indicates a higher conversion of androgens to estrogens, which are important for processes such as oocyte maturation and spawning, i.e. more so for females than for males. Thus, the relatively low intensity of IHC labeling in intact animals (in cells of types 3 and 4) may be associated with their juvenile stage of ontogenesis, as well as male gender. After long-term (Fig. 2) and long-term + repeated (Fig. 3) TBI, the intensity of IHC labeling of ApoB in cells of types 1 and 2 remains moderate/intense (++/+++), which corresponds to the data for cells of this type in intact animals . The difference lies in the number of type 1 cells in intact animals and a significant increase in the number of these cells in all areas of the cerebellum with long-term and long-term + repeated TBI. When IHC labeling of GS, moderate/high intensity of labeling is preserved in cells of types 1 and 2 (Fig. 4A, B), the number of which is small in intact animals. Moderate intensity of IHC labeling was also detected in elongated migrating type 4 cells (Fig. 4D, see Table 2). The background staining intensity of the IHC reaction to HS is low, but cells 1, 2 (Fig. 4A, B, E, D) and type 4 cells (Fig. 4D) are clearly visible. After long-term TBI, cells of types 1 and 2 in the superficial zones (Fig. 5A), the molecular layer and reactive neurogenic niches in the thickness of the molecular layer (Fig. 5C, D, E) also have a moderate/intense intensity of immunolabeling. In the DMZ (Fig. 5F) and granular eminences (Fig. 5G), the intensity of IHC labeling of GS in cells of this type is also intense or moderate. In all studied areas of the cerebellum, the number of type 1 cells significantly increases compared to intact animals (Fig. 5H). With repeated trauma, the intensity of IHC labeling of type 1 and 2 cells is high (Fig. 6A-F). The number of GS+ cells of types 1 and 2 increases in ALL studied areas of the cerebellum (Fig. 6G). When IHC labeling on CBS, cells of types 1 and 2 in intact animals exhibit from weak (Fig. 7A, B) to intense (Fig. 7C) intensity of immunolabeling. After long-term injury, the overall intensity of immunolabeling in cells of types 1 and 2 increases to +/+++ (Fig. 8A, B, D) and the number of such cells significantly increases in different areas of the cerebellum (Fig. 8F). As a result of repeated trauma, the intensity of CBS IHC labeling in type 1 and 2 cells increases to high intensity (Table 3, Fig. 9 A-D) and the number of such cells increases significantly (Fig. 9G). Thus, in this work, the results imply a large, comprehensive analysis of AroB, GS, and CBS-immunopositive cells, which includes several independent stages: a) analysis of morphometric data on the number of immunopositive cells of types 1 and 2, b) densitometric characteristics of immunolabeling and c) directly morphological data on the topography of immunopositive cells in various areas of the cerebellum, which is confirmed by the results presented in the photographs, as well as by statistical analysis data. In accordance with the reviewer's recommendations, the text of the article has been shortened.

Reviewer 2 Report

Comments and Suggestions for Authors

This manuscript describes comparative immunohistochemistry experiments using juvenile Onchorhynchus keta. The focus is on three proteins, aromatase B, glutamine synthetase, and cystathionine-beta-synthase. The work presented is extensive, and appears to be well done. However, the manuscript is overly long and filled with tangential information, and the manuscript is in need of clarification.

- The introduction is long and unfocused. While it includes a great deal of information about the enzymes to be studied in the body of the work, it does not effectively set up the case for why the work needed to be done. In particular, it is not clear why the particular proteins studied were chose, or what their role is in traumatic brain injury. The introduction also barely mentions traumatic brain injury.

- The manuscript discusses Type 1-4 cells extensively, but has barely any detail about what these are. In the context of a more general journal like IJMS, there needs to be more detail about how these types of cells are identified, and what their significance is.

- The tables do a great job of summarizing the histologic findings. It feels odd that the optical density for the various stains, which appears to have been measured quantitatively, is presented as +/++/+++. It seems that it would make more sense to report the numbers here, as well as comparative statistics between them as is done for cell counts. The authors may want to consider changing the column titles for 3 month and 3 month + 1 week to TBI and repeat TBI, to more clearly communicate the significance of these time points.

 - Figures 10, 11, and 12 appear unnecessary – they are merely a combination of graphs from previous figures, with the expression of the three proteins in an additive bar graph. There is not a significant connection drawn between the three proteins that would suggest there is a reason to add the cell counts that are positive for the three into a single bar.

- The discussion is also unfocused, and spends too much writing on information that is only tangentially related to the findings of the current study. In this way it reads more like a comprehensive review than a discussion. An example of this is spending a majority of the last 3 pages of the discussion on hydrogen sulfide. Although CBS is involved in producing H2S, it also has other activities, and there are no results presented that suggest H2S specifically was investigated. Thus, although it would be appropriate to mention H2S as a potential downstream effect of changes in CBS expression, 3 pages is excessive.

- Were the animals used in the experiments male or female? Particularly given the importance of aromatase B in estrogen metabolism, this would seem to be an important piece of information.

- In the methods, I am a little unclear about euthanasia/fixation procedures. The methods state the fish were euthanized by rapid decapitation, but then states that they were anesthetized and perfused. I recommend clarifying what was done and in what order.

- The conclusion gives a brief overview of the data presented, which is useful to the reader. However, I am not clear what the actual conclusions of the research are, unless it is just that levels of aromatase B, glutamine synthase, and CBS are higher after injury.

- The English usage is for the most part good. There are a number of instances of incorrect word usage, however, that in some cases obscures the meaning of the writing. English editing is recommended to make sure the meaning is clearer (e.g. the last sentence of the first paragraph of the introduction, lines 56-57).  Similarly, it is recommended the authors carefully proofread numbers reported; for example, p < 0.5 is mentioned in several places (e.g. lines 367, 369), where from context it appears the authors meant p < 0.05.

Comments on the Quality of English Language

- The English usage is for the most part good. There are a number of instances of incorrect word usage, however, that in some cases obscures the meaning of the writing. English editing is recommended to make sure the meaning is clearer (e.g. the last sentence of the first paragraph of the introduction, lines 56-57).  Similarly, it is recommended the authors carefully proofread numbers reported; for example, p < 0.5 is mentioned in several places (e.g. lines 367, 369), where from context it appears the authors meant p < 0.05.

Author Response

This manuscript describes comparative immunohistochemistry experiments using juvenile Onchorhynchus keta. The focus is on three proteins, aromatase B, glutamine synthetase, and cystathionine-beta-synthase. The work presented is extensive, and appears to be well done. However, the manuscript is overly long and filled with tangential information, and the manuscript is in need of clarification.

- The introduction is long and unfocused. While it includes a great deal of information about the enzymes to be studied in the body of the work, it does not effectively set up the case for why the work needed to be done. In particular, it is not clear why the particular proteins studied were chose, or what their role is in traumatic brain injury. The introduction also barely mentions traumatic brain injury.

In accordance with the reviewer's recommendations, the Introduction section was revised and the role of the enzymes AroB, CBS, GS in traumatic brain injury was indicated. A separate section highlights the study of TBI in fish models (moved to the beginning) and the possibility of using these data in understanding models of TBI in humans.

- The manuscript discusses Type 1-4 cells extensively, but has barely any detail about what these are. In the context of a more general journal like IJMS, there needs to be more detail about how these types of cells are identified, and what their significance is.

Thank you for this comment. Cells of types 1-4 were previously isolated in the cerebellum of juvenile chum salmon and masu salmon in accordance with morpho-functional, topographic and proliferative characteristics (Pushchina, E.V.; Stukaneva, M.E.; Varaksin, A.A. Hydrogen Sulfide Modulates Adult and Reparative Neurogenesis in the Cerebellum of Juvenile Masu Salmon, Oncorhynchus masou. Int. J. Mol. Sci. 2020, 21, 9638. https://doi.org/10.3390/ijms21249638). Previously, a detailed justification was given for the need to isolate these cell types and an indication that type 1 cells correspond to NSCP (Pushchina et al., 2020). Cells of the 2nd type are elongated, larger than cells of the 1st type, and also probably represent the stage of pre-proliferation (cells of the 1st type before division), and may also correspond to mammalian astrocytes (in the discussion, in the section on GS about this it says). Type 3 cells are cells of the ganglion layer and are represented by two populations: Purkinje cells and EDCs. Type 4 cells are an elongated migratory population of cells involved in the reparative process and capable of migrating towards the injury zone.

- The tables do a great job of summarizing the histologic findings. It feels odd that the optical density for the various stains, which appears to have been measured quantitatively, is presented as +/++/+++. It seems that it would make more sense to report the numbers here, as well as comparative statistics between them as is done for cell counts. The authors may want to consider changing the column titles for 3 month and 3 month + 1 week to TBI and repeat TBI, to more clearly communicate the significance of these time points.

Previously, when assessing the densitometric parameters of IP cells, we already used quantitative optical density values (Pushchina, E.V.; Zharikova, E.I.; Varaksin, A.A.; Prudnikov, I.M.; Tsyvkin, V.N. Proliferation, Adult Neuronal Stem Cells and Cells Migration in Pallium during Constitutive Neurogenesis and after Traumatic Injury of Telencephalon of Juvenile Masu Salmon, Oncorhynchus masou. Brain Sci. 2020, 10, 222. https://doi.org/10.3390/brainsci10040222). In accordance with the reviewer's recommendations, absolute optical density values are included in the table.

 - Figures 10, 11, and 12 appear unnecessary – they are merely a combination of graphs from previous figures, with the expression of the three proteins in an additive bar graph. There is not a significant connection drawn between the three proteins that would suggest there is a reason to add the cell counts that are positive for the three into a single bar.

As recommended by the Reviewer, Fig. 10-12 were modified and presented in the form of comparative histograms with an assessment of intragroup differences in intact animals and an assessment of intergroup differences in primary and repeated injuries.

- The discussion is also unfocused, and spends too much writing on information that is only tangentially related to the findings of the current study. In this way it reads more like a comprehensive review than a discussion. An example of this is spending a majority of the last 3 pages of the discussion on hydrogen sulfide. Although CBS is involved in producing H2S, it also has other activities, and there are no results presented that suggest H2S specifically was investigated. Thus, although it would be appropriate to mention H2S as a potential downstream effect of changes in CBS expression, 3 pages is excessive.

In accordance with the recommendations of the Reviewers, the section related to the participation of hydrogen sulfide in the constitutive and traumatic reparative processes in the cerebellum of juvenile chum salmon was significantly shortened in the Discussion section.

- Were the animals used in the experiments male or female? Particularly given the importance of aromatase B in estrogen metabolism, this would seem to be an important piece of information.

Most of the fish were males

- In the methods, I am a little unclear about euthanasia/fixation procedures. The methods state the fish were euthanized by rapid decapitation, but then states that they were anesthetized and perfused. I recommend clarifying what was done and in what order.

The necessary clarifications have been made to this Methods section.

- The conclusion gives a brief overview of the data presented, which is useful to the reader. However, I am not clear what the actual conclusions of the research are, unless it is just that levels of aromatase B, glutamine synthase, and CBS are higher after injury.

The Conclusion section has been edited in accordance with the recommendations. In conclusion, the necessary conclusions are presented and it is shown that as a result of traumatic injury, the number of cells (types 1 and 2) producing aromatase B, glutamine synthase and CBS increases in all areas of the cerebellum of juvenile chum salmon, which indicates the participation of these markers in regenerative neurogenesis. Regarding ApoB, this conclusion is shown for the first time.

- The English usage is for the most part good. There are a number of instances of incorrect word usage, however, that in some cases obscures the meaning of the writing. English editing is recommended to make sure the meaning is clearer (e.g. the last sentence of the first paragraph of the introduction, lines 56-57).  Similarly, it is recommended the authors carefully proofread numbers reported; for example, p < 0.5 is mentioned in several places (e.g. lines 367, 369), where from context it appears the authors meant p < 0.05.

Editing of the English language was carried out throughout the article in accordance with the recommendations made.

Reviewer 3 Report

Comments and Suggestions for Authors

The paper entitled “Comparative Analysis and Post-Traumatic Dynamics of Expression of Aromatase B, Glutamine Synthetase, and Cystathionine-beta-Synthase in the Cerebellum of Juvenile Chum Salmon, Oncorhynchus keta” by Pushchina E.V. and the co-authors is an excellent contribution to highlight the impact of TBI on the expression of Aromatase B, Glutamine Synthetase, and Cystathionine-beta-Synthase in the Cerebellum of Juvenile Chum Salmon. The authors have found that upon repeated injury, the number of neuronal progenitor cells in the juvenile O. keta cerebellum continued to increase without a noticeable decrease in the regenerative potentials.

The paper is a nice contribution to the topic of neurogenesis and may be helpful for studies related to brain injury and neurogenesis.

The comments may be given below.

1.      The title may be revised and maybe simplified.

2.      Why did the authors opt for fish instead of normal rodents?

3.      The abstract last parts need careful trimming to highlight the main outcomes of the study.

4.      The introduction is more than enough. Please remove the unnecessary information.

5. In Figure 2, the methods used to quantify the images are not clear. Please elaborate on it in detail.

6.      The conclusion is not clear and to the point.

7.      Much part of the discussion is just an introduction.

Comments on the Quality of English Language

 Minor editing of the English language required

Author Response

The paper entitled “Comparative Analysis and Post-Traumatic Dynamics of Expression of Aromatase B, Glutamine Synthetase, and Cystathionine-beta-Synthase in the Cerebellum of Juvenile Chum Salmon, Oncorhynchus keta” by Pushchina E.V. and the co-authors is an excellent contribution to highlight the impact of TBI on the expression of Aromatase B, Glutamine Synthetase, and Cystathionine-beta-Synthase in the Cerebellum of Juvenile Chum Salmon. The authors have found that upon repeated injury, the number of neuronal progenitor cells in the juvenile O. keta cerebellum continued to increase without a noticeable decrease in the regenerative potentials.

The paper is a nice contribution to the topic of neurogenesis and may be helpful for studies related to brain injury and neurogenesis.

The comments may be given below.

  1. The title may be revised and maybe simplified.

In accordance with the recommendations made, we simplified the title of the work.

  1. Why did the authors opt for fish instead of normal rodents?

The study of the processes of neurogenesis in juvenile chum salmon is interesting because it is an endemic species. In addition to studying the characteristics of the restoration of the central nervous system of a valuable commercial species, the chum salmon population is easily accessible for study by our team, which allows us to conduct research at the required age stages. Also, phylogenetically ancient groups, which include salmonids, are characterized by a high concentration of undifferentiated elements both in the matrix areas of the brain and in the parenchyma. In adulthood, chum salmon retains the embryonic state of individual organs or their systems, which include the central nervous system, due to embryonication processes. It is noteworthy that these processes are superimposed on the stage of active growth, when intensive processes of neurogenesis occur, which makes this species a convenient model for research. A promising strategy for the development of effective cell replacement therapy, involving the study of organisms capable of regeneration, will give us a unique opportunity to gain a broad understanding of the restoration of tissues of the adult central nervous system. Thus, the study of regenerative-competent organisms, in particular juvenile Pacific salmon, can clarify how regeneration occurs in species with delayed embryonic developmental signs (fetalization) and what role the enzymes previously discovered in the NSPC of juvenile salmon, in particular GS and CBS, play. as well as radial glia of the aNSPC of other fish.

  1. The abstract last parts need careful trimming to highlight the main outcomes of the study.

The necessary changes were made to the annotation in accordance with the recommendations made.

  1. The introduction is more than enough. Please remove the unnecessary information.

The Introduction has been thoroughly revised and abbreviated in accordance with the recommendations

  1. In Figure 2, the methods used to quantify the images are not clear. Please elaborate on it in detail.

Necessary notation has been added to Figure 2 to clarify data quantification methods.

  1. The conclusion is not clear and to the point.

Significant changes and clarifications have been made to the Conclusion, confirming the most interesting conclusions of the work.

  1. Much part of the discussion is just an introduction.

A thorough revision was carried out in the Discussion and reductions were made in the section related to hydrogen sulfide.

Round 2

Reviewer 1 Report

Comments and Suggestions for Authors

The quality of images/specimens is very poor and it has not be improved. The Results (images) do not support the conclusions of the manuscript.

The manuscript, as it is, cannot be published in a scientific journal.

Author Response

In our experiment, when analyzing the data obtained, the low aromatase activity in the intact brain of juvenile chum salmon is explained by the fact that the fish population was represented mainly by males, who are characterized by low levels of estrogen. When assessing the dynamics of reparative processes: with long-term injury and repeated acute injury, the montages clearly show an increase in the intensity of marking, structural restructuring consisting in the formation of neurogenic niches and a general significant increase in the number of cells, which is consistently reflected in the chapters “results” and “discussion”. The data we obtained are the results of a pure experiment without adjustment to the expected result. In addition, this article is part of a series of sequential works on a comprehensive study of the processes of neurogenesis in Pacific salmon, which were previously published in the journal IJMS, have been well viewed and cited by colleagues in the field.

Unfortunately, Reviewer 1 did not indicate specific comments on each installation, limiting himself to short sentences stating that the data obtained did not satisfy him. If we redo the article in accordance with the wishes of Reviewer 1, then it will be a new article, with new data and their interpretation. In this case, the work of the other two reviewers, who were generally positive about our work after careful analysis, would be lost. We carefully corrected all the comments that were listed point by point in accordance with the recommendations of reviewers 2 and 3, as evidenced by the absence of comments in the second round of review from reviewer 3 and small suggestions from reviewer 2, which we eliminated.

Reviewer 2 Report

Comments and Suggestions for Authors

The revisions significantly improve the submitted manuscript. The authors have addressed most of my concerns, with only a couple minor clarifications needed.

- The information that most of the fish in the study were males should be added to the methods section under 4.1 Experimental Animals.

- The information contained in the response about the Type 1 and 2 cells does not appear to have been added to the paper. I would still recommend adding this, because IJMS is a more general audience journal, and it is likely that many readers are not necessarily accustomed to histology of teleost fish brains, and thus are unlikely familiar with this terminology. This could easily be added to the methods section describing how the cell types are defined and what they represent.

- In the final paragraph of the conclusion, I recommend softening the first sentence. Although based on the functions of the studied proteins it is possible or even likely that they are playing neuroprotective roles, this was not directly assessed in the manuscript. Thus, it would be more correct to word the sentence as “thus, the markers we discovered in the cerebellum of juvenile chum salmon may act as neuroprotectors..” or “likely act as neuroprotectors…”

Author Response

- The information that most of the fish in the study were males should be added to the methods section under 4.1 Experimental Animals.

 Information has been added to section 4.1 Experimental Animals.

- The information contained in the response about the Type 1 and 2 cells does not appear to have been added to the paper. I would still recommend adding this, because IJMS is a more general audience journal, and it is likely that many readers are not necessarily accustomed to histology of teleost fish brains, and thus are unlikely familiar with this terminology. This could easily be added to the methods section describing how the cell types are defined and what they represent.

Information has been added to the Results section.

​- In the final paragraph of the conclusion, I recommend softening the first sentence. Although based on the functions of the studied proteins it is possible or even likely that they are playing neuroprotective roles, this was not directly assessed in the manuscript. Thus, it would be more correct to word the sentence as “thus, the markers we discovered in the cerebellum of juvenile chum salmon may act as neuroprotectors..” or “likely act as neuroprotectors…”

Corresponding corrections have been made